# DISTILL VISUAL CHART REASONING ABILITY FROM LLMS TO MLLMS

## ABSTRACT

Solving complex chart Q&A tasks requires advanced visual reasoning abilities in multimodal large language models (MLLMs). Recent studies highlight that these abilities consist of two main parts: recognizing key information from visual inputs and conducting reasoning over it. Thus, a promising approach to enhance MLLMs is to construct relevant training data focusing on the two aspects. However, collecting and annotating complex charts and questions is costly and time-consuming, and ensuring the quality of annotated answers remains a challenge. In this paper, we propose *Code-as-Intermediary Translation* (CIT), a cost-effective, efficient and easily scalable data synthesis method for distilling visual reasoning abilities **from LLMs to MLLMs**. The code serves as an intermediary that translates visual chart representations into textual representations, enabling LLMs to understand cross-modal information. Specifically, we employ text-based synthesizing techniques to construct chart-plotting code and produce **REACHQA**, a dataset containing 3k reasoning-intensive charts and 20k Q&A pairs to enhance both recognition and reasoning abilities. Experiments show that when fine-tuned with our data, models not only perform well on chart-related benchmarks, but also demonstrate improved multimodal reasoning abilities on general mathematical benchmarks such as MathVista.

## 1 INTRODUCTION

Multimodal large language models (MLLMs) have made significant achievements, particularly in visual recognition tasks (OpenAI, 2024a; Anthropic, 2024). While they can efficiently handle simple visual inputs, there has been a growing emphasis on complex chart understanding, driven by the widespread use of charts in real-world contexts (Masry et al., 2022; Huang et al., 2024). However, addressing reasoning-intensive questions involving charts remains challenging for these models. Existing benchmarks underscore the need for more advanced and generalized visual reasoning abilities, which remain underdeveloped in current MLLMs (Wang et al., 2024c; Lu et al., 2024). In contrast, humans excel in such tasks since they can purposefully identify query-relevant information from images and engage in step-by-step reasoning (Wang et al., 2024c;a). These behaviors offer insights into how advanced visual reasoning could be modeled. Our analysis of the error distribution on ChartQA (Masry et al., 2022) in Figure 1 also highlights two main types of model failure. Specifically, 62% of errors occur during the initial recognition phase, while 36% arise from reasoning mistakes following correct recognition. It shows that even advanced MLLMs struggle with basic recognition and make superficial errors during reasoning. In light of these findings, learning to solve problems in a human-like manner becomes essential for current models.

One promising strategy is to distill expert trajectories, such as those annotated by human experts or stronger models (Han et al., 2023; Meng et al., 2024; Masry et al., 2024a;b). However, constructing such high-quality trajectories for complex chart tasks is a costly and time-consuming process. The challenges arise from both the chart and the instruction data. Collecting charts from specific online sources typically involves manual crawling and filtering, followed by extensive human annotation to create relevant questions (Masry et al., 2022; Wang et al., 2024c). While some approaches use LLMs to automate Q&A synthesis, they rely on the data table of charts (Han et al., 2023; Masry et al., 2024a), which neglects features like color, layout, and structure critical for visual recognition. Even when MLLMs are used for Q&A generation (Masry et al., 2024b), their performance remains suboptimal. Our preliminary analysis (§ 2.2) further indicates that proprietary MLLMs like GPT-

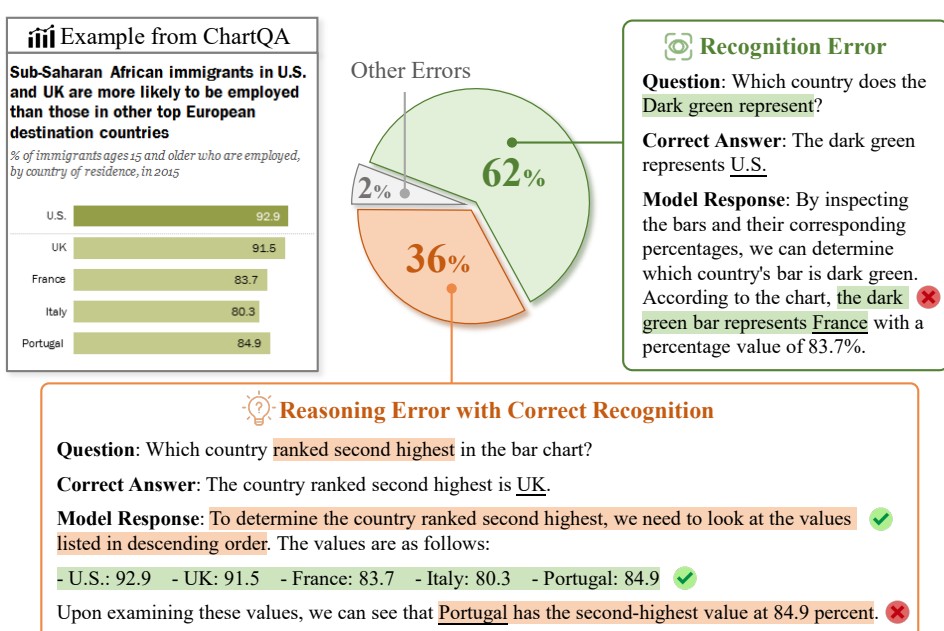

Figure 1: Error distribution of incorrect answers by MiniCPM-V2.5-Llama3 (Yao et al., 2024) on the ChartQA testing set, as judged by GPT-4o. We present an example chart from ChartQA along with two error cases: one for recognition and one for reasoning. The "Other Errors" include question misunderstood errors, knowledge and hallucination errors, or refusal to answer.

4o also encounter difficulties in producing accurate and challenging chart-related Q&A pairs. In contrast, LLMs tend to perform better when processing charts in textual format, demonstrating lower costs and higher accuracy, as also observed by Fu et al. (2024).

Given the limitations of MLLMs in this task, we propose leveraging the strengths of LLMs through an intermediary representation: **code**. Inspired by the concept of intermediary translation (Zarechnak, 1986; Léon, 2007), which refers to using a bridge language to improve translation quality across diverse languages in literary studies, we introduce **Code-as-Intermediary Translation (CIT)**. The code acts as an intermediary, translating chart images into textual representations while preserving visual features faithfully. This process enables LLMs to interpret cross-modal information and generate more accurate, visually complex Q&A pairs. Furthermore, it facilitates the adoption of text-based instruction augmentation strategies, such as Self-Instruct (Wang et al., 2023) and Evol-Instruct (Xu et al., 2024), to enhance the diversity and complexity of the generated charts. Starting with 33 seed codes collected from the Matplotlib gallery, we synthesize more chart-plotting codes covering diverse types and topics, and then complicate them to create richer ones. Finally, using the synthetic codes as a bridge, we generate charts (via Python) and instructions (via LLMs) in a bi-directional process, ensuring the alignment between the two modalities.

With the CIT approach, we construct **REACHQA**, a multimodal instruction dataset containing 3, 249 **rea**soning-intensive **ch**arts and 19, 963 Q&A pairs, all at a remarkably low cost of just $300. The dataset comprises 8k questions focused on visual recognition and 12k on reasoning, designed to address the dual challenges of extracting visual information and performing advanced reasoning. Additionally, we create 500 charts and 2, 000 manually verified Q&A pairs to independently assess models' recognition and reasoning abilities. When fine-tuning with REACHQA, all models demonstrate substantial performance gains across benchmarks, with LLaVA-Next-Llama3-8B (Li et al., 2024) improving by over 30% on average. More importantly, we observe that these gains generalized beyond chart-specific tasks to broader multimodal reasoning tasks, such as MathVista and MATH-Vision—an outcome previously unattainable with existing chart-focused datasets. Furthermore, we conduct experiments to investigate deeper insights, including (1) the real impact of expert trajectories on reasoning abilities, (2) the effects of recognition- and reasoning-oriented training data ratios, and (3) the benefits of mixing general-purpose multimodal instruction data.

Table 1: Comparison of existing chart-related datasets across three properties. Only the chart question-answering (CQA) task is considered, despite some datasets having multiple tasks. Abbreviations: Vis.=visual, Comp.=complexity, Temp.=template, Refer.=Reference, Reas.=reasoning, Rat.=rationale, Annot.=annotation and Scal.=scalable. Cells marked with "✓" indicate mixed attributes (e.g., partially template-based; scalable Q&A but non-scalable charts.). "*" indicates that while the chart-plotting codes are public, the Q&A generation still relies on data tables.

| Datasets | Chart Properties | | | | Q&A Properties | | | Dataset Properties | | |
|---|---|---|---|---|---|---|---|---|---|---|
| | # Chart Type | # Chart Topic | Textual Format | Vis. Comp. | Temp. Free | Vis. Refer. | Rat. Annot. | Train Set | Test Set | Scal. |
| PlotQA (Methani et al., 2020) | 3 | - | Table | ✗ | ✗ | ✓ | ✗ | ✓ | ✓ | ✗ |
| ChartQA (Masry et al., 2022) | 3 | 15 | Table | ✗ | ✓ | ✓ | ✗ | ✓ | ✓ | ✗ |
| OpenCQA (Kantharaj et al., 2022) | 5 | 10 | Caption | ✗ | ✓ | ✗ | ✓ | ✗ | ✓ | ✗ |
| MathVista (Lu et al., 2024) | - | - | - | ✗ | ✓ | ✗ | ✗ | ✗ | ✓ | ✗ |
| CharXiv (Wang et al., 2024c) | - | - | - | ✓ | ✓ | ✓ | ✗ | ✗ | ✓ | ✗ |
| ChartBench (Xu et al., 2023) | **9 / 42** | - | Table | ✗ | ✗ | ✗ | ✗ | ✓ | ✓ | ✓ |
| ChartX (Xia et al., 2024) | 18 | 22 | Code* | ✗ | ✓ | ✗ | ✗ | ✗ | ✓ | ✓ |
| MMC (Liu et al., 2024a) | 6 | 5 | Caption | ✓ | ✓ | ✗ | ✓ | ✓ | ✓ | ✓ |
| ChartLlama (Han et al., 2023) | 10 | - | Table | ✗ | ✓ | ✗ | ✓ | ✓ | ✓ | ✓ |
| ChartAst (Meng et al., 2024) | 9 | - | Table | ✗ | ✗ | ✗ | ✓ | ✓ | ✗ | ✓ |
| ChartInstruct (Masry et al., 2024a) | - | - | Table | ✗ | ✓ | ✗ | ✓ | ✓ | ✗ | ✓ |
| ChartGemma (Masry et al., 2024b) | - | - | - | ✗ | ✓ | ✓ | ✓ | ✓ | ✗ | ✓ |
| **REACHQA (ours)** | **10 / 32** | ∞ | **Code** | ✓ | ✓ | ✓ | ✓ | ✓ | ✓ | ✓ |

## 2 BACKGROUND

### 2.1 DEFICIENCIES IN EXISTING CHART-RELATED DATASETS

Existing chart-related datasets fall into two categories: benchmarks for evaluation and training data to improve model performance. These datasets are either collected from online data sources or generated by LLMs, sometimes requiring manual annotation or automated question generation. Most of them focus on basic visual recognition tasks. Though some recent works target more advanced reasoning, they often lack scalability. Table 1 summarizes these datasets, with further details below.

**Chart Properties.** The *visual diversity* is shaped by the variety of chart types and topics (Wang et al., 2024c). Early datasets like ChartQA and OpenCQA (Kantharaj et al., 2022), sourced from limited websites, featured uniform styles with minimal diversity. To address this, recent works like ChartAst (Meng et al., 2024) synthesize charts with randomized attributes (e.g., color, fonts) using LLMs. However, beyond the superficial variations in chart appearance, many of them overlook the *visual complexity* (Zeng et al., 2024). As models evolve, simple style changes no longer pose challenges. Datasets like CharXiv and MMC (Liu et al., 2024a), which include complex scientific charts from arXiv papers, naturally exhibit greater complexity in recognition. Additionally, the textual format of charts is critical, as it can be used to expand the datasets via LLMs.

**Q&A Properties.** Some benchmarks like PlotQA and ChartBench (Xu et al., 2023) use predefined templates to generate Q&A pairs, resulting in monotonous and simplistic questions. Other datasets, such as ChartQA and CharXiv, required manual annotation, which improved quality but increased costs and hindered scalability. With the advent of LLMs, works like ChartLlama (Han et al., 2023) and ChartInstruct (Masry et al., 2024a) use them to generate diverse questions from data tables while also providing rationale annotations for training. However, these methods fail to capture fine-grained visual elements like color, layout, and structure because they rely on only the data. To address this, ChartGemma (Masry et al., 2024b) uses MLLMs to generate Q&A pairs directly from charts.

**Dataset Properties.** While manually annotated datasets like MathVista (Lu et al., 2024) and CharXiv provide high-quality data, their development is resource-intensive, typically resulting in datasets of only a few thousand samples. In the era of LLMs, such methods are impractical for scaling to the size needed for training large models. Recent efforts, such as ChartAst, ChartInstruct, and ChartGemma, have explored Q&A generation for dataset expansion, but they remain limited by the difficulty of collecting a large set of charts. A more scalable approach is to leverage the generative capabilities of LLMs to synthesize charts like ChartBench and ChartX (Xia et al., 2024).

## 2.2 CAN LLMS UNDERSTAND CHARTS WITHOUT VISUAL INPUT?

To explore whether there is a more effective textual format for representing visual information than data tables, we propose using code. Code, which can precisely encode chart structures, may serve as an ideal bridge between modalities. We design an experiment to test this hypothesis.

We collect 25 complex charts, along with their corresponding data tables and code, from research papers by graduate students in the college. These charts often feature multiple or overlay plots and dense data groups, with the code averaging over 100 lines. Such content is unlikely to be included in the training data of current models. For each sample, GPT-4o receives three types of input—table, code, and chart images—to generate a challenging Q&A pair. In total, 75 pairs are created, randomly shuffled, and then presented to three annotators for blind evaluation.

The annotators are asked to rate each pair on accuracy, reasoning complexity, and visual reference, using a scale of 1 (low) to 3 (high). The results in Table 2 indicate that both text-based inputs outperform chart input in the first two aspects, with code scoring 2.60 in accuracy (vs. 1.91) and 2.56 in reasoning complexity (vs. 1.53). As expected, table input has the lowest visual reference score (1.19), while chart input scores highest in this (2.36), confirming the ability of MLLMs to directly interpret visual information. Surprisingly, the code achieves a relatively high visual reference score (2.15) even without image input. It suggests that code has the potential to translate visual charts into textual representations.

Table 2: Rating results and costs for different input types in our study.

| Input | Acc. | Reas. Comp. | Vis. Refer. | Cost ($) |
|-------|------|-------------|-------------|----------|
| Table | **2.72** | 2.51 | 1.19 | **0.047** |
| Code | 2.60 | **2.56** | 2.15 | 0.092 |
| Chart | 1.91 | 1.53 | **2.36** | 0.107 |

## 3 REACHQA: SYNTHESIZING CHART Q&A WITH CIT

REACHQA is a multimodal Chart Question-Answering (CQA) dataset fully generated by LLMs, consisting of synthetic charts paired with corresponding Q&A. It is constructed with Code-as-Intermediary Translation (CIT), a data synthesis method for distilling visual reasoning abilities from LLMs to MLLMs, as illustrated in Figure 2. In the following sections, we describe how we synthesize the codes (§ 3.1), generate charts and instructions (§ 3.2), and ensure data quality (§ 3.3).

### 3.1 INTERMEDIARY CODE SYNTHESIS

This section will describe how the intermediary code is synthesized. The prompt templates are detailed in Appendix H.1.

**Seed Code Collection.** We start by collecting a small set of 33 seed code samples, which we refer to as $C_{\text{seed}}$. These samples are sourced directly from the official Matplotlib gallery[1] to ensure quality and minimize manual effort. Collectively, the code samples, each averaging around 40 lines in length, cover a diverse range of chart types, including common types like bar, line, and scatter charts, as well as more specialized charts such as bubble, contour, and donut charts. All samples are verified for executability to guarantee the reliability of the subsequent code synthesis process.

**Self-Instruct for Diverse Code Generation.** To expand the diversity and coverage of the synthetic chart set, we apply the Self-Instruct method (Wang et al., 2023), which has been used to generate new instructions by presenting LLMs with existing ones. Plenty of works have demonstrated its effectiveness (Liu et al., 2024b). In our approach, instead of instructions, we provide code samples as context, guiding the model to generate novel chart-plotting code. At each step, 3 randomly selected code snippets from the code pool serve as few-shot examples (Brown et al., 2020).

To diversify chart generation, a chart type is randomly chosen from 10 major and 32 minor categories for the model to synthesize. For chart content, we provide two topic options, allowing the model to freely combine or expand on these themes based on its knowledge, leading to varied topics and data. A chain-of-thought (CoT) process (Wei et al., 2022) is used for code generation, starting with

---

[1]https://matplotlib.org/stable/gallery/index.html

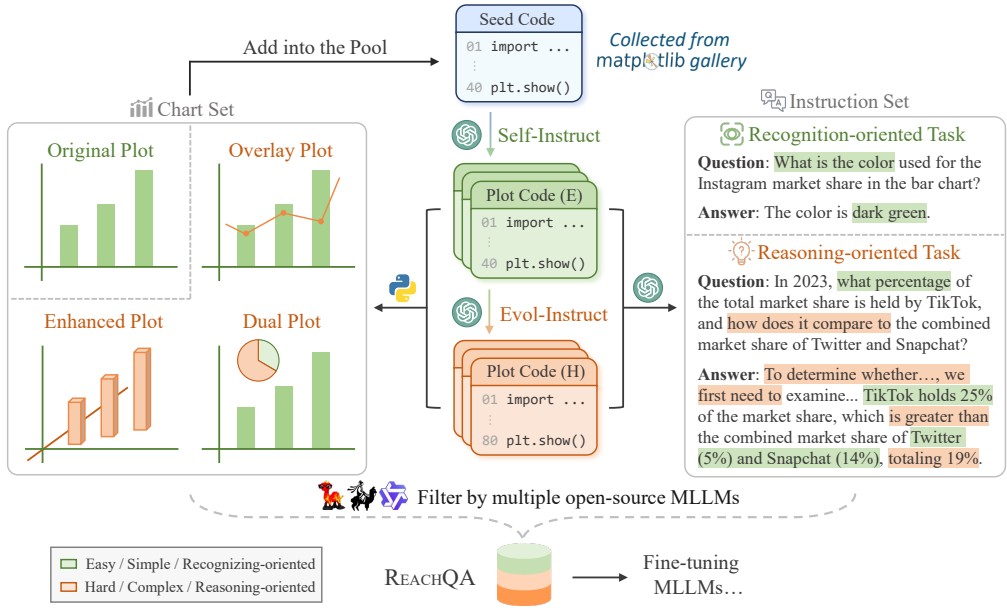

Figure 2: Overview of the Code-as-Intermediary Translation (CIT) method for synthesizing multimodal instruction data. The process begins with 33 seed codes and generates plot codes across various chart types, topics, and complexity levels through the Self-Instruct and Evol-Instruct stages. The chart set and instruction set are constructed bi-directionally, and the final filtered data yields REACHQA, a dataset for distilling visual chart reasoning abilities from LLMs to MLLMs.

the chart's background and data, followed by the final executable code. This step-by-step approach ensures logical coherence and code functionality. The generated codes are referred to as $C_{\text{easy}}$ for use in subsequent phases of the construction. The chart types and topics are detailed in Appendix A.

**Evol-Instruct for Complex Code Generation.** To enhance the complexity of the synthetic chart set, we adopt the Evol-Instruct method (Xu et al., 2024). Evol-Instruct leverages LLMs to evolve simple chart-plotting code into more intricate versions by presenting existing code alongside an evolution strategy as context. This approach addresses a key limitation in prior work that emphasizes the quantity of charts while often neglecting the difficulty of chart interpretation.

At each step, we take code samples from $C_{\text{easy}}$ and evolve them with a randomly selected strategy from four predefined directions:

1. Expanding the data size or number of data groups;
2. Adding or modifying visual elements to enhance presentation;
3. Overlaying a different type of chart on the original plot;
4. Introducing an additional subplot beside the original plot.

These strategies ensure that the resulting charts demand more nuanced visual interpretation and indepth reasoning. As in previous phases, we follow a CoT process, where the model first analyzes the given code, outlines the evolution details, and then generates the evolved code. The final codes, referred to as $C_{\text{hard}}$, are added to the code pool for subsequent use, too.

## 3.2 BI-DIRECTIONAL TRANSLATION

This section outlines the construction process of charts and instruction sets. The prompt templates are detailed in Appendix H.2.

**Chart Generation through Code Execution and Self-Repair.** We generate charts by executing the Python plotting code. However, during the generation and evolution process, program errors

are inevitable. To ensure correctness, we will validate the code before adding it to the pool. When errors occur, the code is not immediately discarded; instead, we apply a Self-Repair method (Chen et al., 2024a), feeding the code and execution results into the LLMs for correction. This process repeats until the code is fixed or reaches an iteration limit, after which the code is discarded if it remains faulty. On average, this approach fixes about 15% of the code generated by GPT-4o, with 5% remaining unrepairable and filtered out.

**Instruction Generation through Guided Prompting.** After verifying the executability of codes, we use them to create instruction sets in the form of question-answer pairs. Building on prior work of in-context Q&A generation (Chen et al., 2023; He et al., 2024), we guide the model through a two-step process: first synthesizing a batch of questions and then generating corresponding answers separately. To ensure high-quality answers, we also employ a step-by-step approach where the model first generates detailed calculations and analyses, which are subsequently refined into concise, educational answers optimized for learning (Gunasekar et al., 2023).

The model will generate two types of instructions: recognition-oriented, focusing on visual information retrieval, and reasoning-oriented, requiring both comprehensive recognition and multi-step reasoning. With minimal constraints on question content, the model is encouraged to explore creative and diverse instructions. Each chart can be used to generate multiple questions, and we filter redundant ones using ROUGE-L overlap, following the Self-Instruct method (Wang et al., 2023).

## 3.3 QUALITY ASSURANCE

This section aims to improve the quality of annotated data. The prompt templates are shown in Appendix H.3.

**Multimodal Validation for Enhanced Data Quality.** Although our dataset is synthesized entirely with LLMs, we recognize the need to integrate visual information for higher data quality (Masry et al., 2024b; Zeng et al., 2024). Therefore, we introduce a multimodal validation step, utilizing MLLMs to verify both generated charts and their corresponding instructions. Due to the variations in model architectures, visual encoders, and training recipes, different models may focus on varying aspects of the images. Taking this into account, we adopt an "majority voting" strategy using multiple open-source models such as MiniCPM-V2.5-Llama3 (Yao et al., 2024), InternVL2-8B (Chen et al., 2024b) and Qwen2-VL-8B (Wang et al., 2024b). This ensures reliable visual validation while remaining cost-effective by ensembling smaller, locally hosted models.

For chart validation, we define a set of criteria for a "good chart", and each model assigns a score between 1 and 5. The average score is calculated, and

Table 3: REACHQA dataset statistics. Question and answer lengths are calculated based on the GPT-4o tokenizer.

| Statistics | Train Set | Test Set |
|---|---|---|
| Total charts | 3,249 | 500 |
| - # Chart types | 10 / 32 | 10 / 32 |
| - # Overlay plots | 1,030 | 220 |
| - # Multiple plots | 593 | 251 |
| - Average size (px) | 2480×1571 | 2798×1601 |
| Unique questions | 19,963 | 2,000 |
| - # Reco. per chart | 2.53 | 2 |
| - # Reas. per chart | 3.62 | 2 |
| Avg. Reco. Q. length | 22.1 | 21.0 |
| Avg. Reco. A. length | 38.3 | 7.0 |
| Avg. Reas. Q. length | 38.2 | 35.4 |
| Avg. Reas. A. length | 68.4 | 24.9 |

charts falling below a threshold are filtered out. For instructions, both the Q&A pairs and the associated charts are fed into the models, which verify the answers step-by-step and give a yes/no judgment. Samples receiving multiple negative votes are discarded.

**Testing Set Construction and Annotation Refinement.** For the REACHQA testing set, we follow a similar process as in previous data generation, but with stricter filtering criteria to ensure higher quality. In addition, annotators are recruited for manual review and refinement. All had at least a bachelor's degree and are familiar with common chart types. They also had prior experience in data annotation. For the charts, annotators first review the images to identify any potential visual errors. For the Q&A pairs, they ensure the questions are relevant to the chart and answerable, then correct any hallucinations or logical inconsistencies in the answers. Afterwards, annotators conduct two rounds of review to confirm the data meets the multimodal recognition or reasoning standards

in our settings. Only samples where at least two annotators agree are accepted into our dataset. The inter-annotator agreement, reflected by a kappa coefficient of 0.82, demonstrates strong consistency (Landis, 1977). Table 3 summarizes the final dataset statistics.

The total cost, excluding open-source model usage and annotation labor for the testing set, was about $300. A detailed expense breakdown is provided in Appendix B.

## 4 EXPERIMENTS

### 4.1 EXPERIMENTAL SETUPS

**Benchmarks.**   We evaluate the models on three categories of tasks. (1) Traditional chart-related benchmarks focused on recognition tasks, including ChartQA (Masry et al., 2022), ChartBench (Xu et al., 2023), and ChartX (Xia et al., 2024). For ChartQA, we use its 2.5k test set. For multi-task datasets like ChartBench and ChartX, we only use QA tasks. Specifically, from ChartBench, we select 2k binary QA samples (Yes/No answers) and 2.1k numerical QA samples. From ChartX, we use its 6k QA samples. (2) Novel chart-related benchmarks that assess both recognition and reasoning abilities, including CharXiv (Wang et al., 2024c) and our REACHQA test set. For CharXiv, we use the validation set, which contains 4k descriptive questions and 1k reasoning questions. Our REACHQA test set includes 1k recognition-oriented and 1k reasoning-oriented questions. (3) General multimodal reasoning benchmarks, including MathVista (Lu et al., 2024) and MATH-Vision (Wang et al., 2024a). For MathVista, we use the testmini set with 540 math-targeted questions and 460 general VQA questions. For MATH-Vision, we use its 3,040 math competition problems.

**Models and baselines.**   We evaluate a range of multimodal large language models (MLLMs) across three categories. (1) Powerful proprietary models, including GPT-4o (OpenAI, 2024a), GPT-4o mini (OpenAI, 2024b), and Claude 3.5 Sonnet (Anthropic, 2024). (2) Chart-augmented open-source models, such as ChartInstruct-7B (Masry et al., 2024a), ChartAssistant-13B (Meng et al., 2024), and ChartGemma-3B (Masry et al., 2024b), which are specifically enhanced for chart-related tasks. (3) Latest general open-source models, including LLaVA-Next-Llama3-8B (Li et al., 2024), MiniCPM-V2.5-Llama3-8B (Yao et al., 2024), and InternVL2-8B (Chen et al., 2024b). Besides, we provide a text-only baseline, denoted as Random (GPT-4o), where we prompt GPT-4o to reasonably guess the answer without seeing the image following Wang et al. (2024c).

**Experiment Details.**   For each general open-source model, we conduct supervised fine-tuning (SFT) using our REACHQA training set. We fine-tuned three versions of each model: (1) using 8k recognition-oriented Q&A samples (denoted as Reco.), (2) using 12k reasoning-oriented samples (denoted as Reas.), and (3) a combined version with 20k samples (denoted as All). We apply Low-rank Adapters (LoRA, Hu et al., 2022) to all linear layers of the language model and projector, with a LoRA rank of 16, a LoRA alpha of 8 and a learning rate of 2e-5.

**Evaluation Metric.**   To fully leverage the capabilities, we evaluate all models using a zero-shot CoT prompt, "Let's think step by step" (Kojima et al., 2022), following OpenAI (2024a) and Anthropic (2024). Thus, to extract answers from the model responses and assess their correctness, we employ the LLM-as-a-judge method to calculate relaxed accuracy (Zheng et al., 2023). The judge model used is GPT-4o, and the prompt template for evaluation can be found in Appendix H.4.

### 4.2 EXPERIMENTAL RESULTS

Table 4 presents the quantitative results for all models and baselines across each task, allowing for a clear comparison of their recognition and reasoning abilities. We can find that:

**Synthetic datasets can also effectively measure abilities.**   Our REACHQA test set effectively evaluates models' reasoning and recognition skills, showing trends similar to human-annotated datasets like CharXiv. For instance, GPT-4o exhibits a reasoning score of 39.70 and a recognition score of 66.80 on REACHQA, closely mirroring its performance on CharXiv (i.e., 47.10 and 84.45, respectively). This consistency suggests that LLM-generated datasets, with minimal human intervention, can rival human-labeled data. Moreover, REACHQA presents a significant challenge

Table 4: Evaluation results on seven benchmarks. Details for these benchmarks and models are presented in § 4.1. The best performance for each category and task is in **bold**. The percentage of performance improvements compared to the vanilla model is denoted by (↑).

| Models | Avg. (↑) | ChartQA QA | ChartBench Binary | ChartBench NQA | ChartX QA | REACHQA Reas. | REACHQA Reco. | CharXiv Reas. | CharXiv Desc. | MathVista Math | MathVista General | MATH-V QA |
|---|---|---|---|---|---|---|---|---|---|---|---|---|
| | | | | | **Baselines** | | | | | | | |
| Human | - | - | - | - | - | 65.10 | 84.60 | 80.50 | 92.10 | 60.30 | | 75.66 |
| Random (GPT-4o) | 20.82 | 30.04 | 40.21 | 22.73 | 19.85 | 8.20 | 13.30 | 10.80 | 19.85 | 17.90 | | 25.36 |
| | | | | **Proprietary Multimodal Large Language Models** | | | | | | | | |
| GPT-4o mini | 49.34 | 77.52 | 70.26 | 34.93 | 35.45 | 27.20 | 53.50 | 34.10 | 74.92 | 56.70 | | 28.85 |
| GPT-4o | 59.85 | 85.70 | **81.03** | **52.88** | 46.60 | 39.70 | 66.80 | 47.10 | **84.45** | 63.80 | | 30.39 |
| Claude 3.5 Sonnet | **64.50** | **90.80** | 76.72 | 48.29 | **58.24** | **51.70** | **74.30** | **60.20** | 84.30 | **67.70** | | **32.76** |
| | | | | **Chart-augmented Multimodal Large Language Models** | | | | | | | | |
| ChartInstruct-7B | 25.93 | 66.64 | 61.40 | 26.95 | 26.62 | 6.00 | 10.50 | 8.80 | **21.40** | 15.37 | 31.52 | **10.07** |
| ChartAssistant-13B | 28.25 | 79.90 | 58.15 | 24.62 | 23.20 | **10.70** | 19.60 | 11.70 | 16.93 | 17.78 | **39.57** | 8.55 |
| ChartGemma-3B | **33.08** | **80.16** | **78.90** | **34.10** | **35.15** | 9.20 | **27.80** | **12.50** | 21.30 | **19.07** | 38.04 | 7.70 |
| | | | | **Open-Source Multimodal Large Language Models** | | | | | | | | |
| LLaVA-Next-Llama3-8B | 24.46 | 45.80 | 42.90 | 15.86 | 15.45 | 6.50 | 17.90 | 17.20 | 31.45 | 22.41 | 44.13 | 9.44 |
| + REACHQA (Reco.) | 32.88 (+34.4%) | **66.96** | 56.95 | **29.52** | 27.25 | 8.80 | 29.00 | 22.20 | 32.58 | 27.40 | 49.78 | 11.25 |
| + REACHQA (Reas.) | 32.39 (+32.4%) | 64.48 | 56.80 | 25.14 | 25.90 | 8.40 | 26.30 | **22.70** | **35.67** | 28.89 | **50.65** | **11.38** |
| + REACHQA (All) | 32.98 (+34.8%) | 64.56 | **57.00** | 29.33 | 27.08 | **11.10** | **29.60** | 22.50 | 32.33 | 27.59 | 50.43 | 11.25 |
| MiniCPM-V2.5-Llama3 | 33.39 | 66.92 | 48.90 | 22.29 | 23.72 | 10.30 | 25.30 | 22.00 | 46.20 | 37.22 | 53.04 | 11.45 |
| + REACHQA (Reco.) | 38.62 (+15.7%) | 71.12 | **56.65** | 33.29 | 29.53 | 10.60 | 34.10 | 25.60 | **48.75** | 41.48 | **60.43** | 13.22 |
| + REACHQA (Reas.) | 38.52 (+15.4%) | 71.72 | **56.65** | 29.62 | 28.23 | **11.00** | 33.00 | 27.50 | 48.70 | **43.52** | 60.22 | 13.52 |
| + REACHQA (All) | 38.67 (+15.8%) | 71.44 | 55.80 | 30.43 | 29.68 | **11.00** | **35.10** | 28.30 | 47.62 | 42.22 | 60.00 | **13.75** |
| InternVL2-8B | 40.03 | 73.80 | 52.05 | 32.86 | 35.10 | 16.20 | 33.70 | 26.30 | 46.10 | 46.11 | 61.74 | 16.38 |
| + REACHQA (Reco.) | 48.21 (+20.4%) | **82.92** | 66.35 | 46.14 | **46.62** | 19.90 | 49.50 | 32.20 | 54.38 | 47.96 | **67.61** | 16.78 |
| + REACHQA (Reas.) | 47.87 (+19.6%) | 82.84 | 64.05 | 46.52 | 44.88 | 20.10 | 49.40 | **32.80** | 52.40 | 49.44 | 66.52 | **17.66** |
| + REACHQA (All) | **48.35 (+20.8%)** | 82.44 | 65.90 | **47.29** | 45.38 | **21.30** | 49.80 | 32.70 | **54.83** | 48.89 | 66.30 | 17.01 |

to models' visual abilities, as random guessing results in very low scores. In contrast, traditional benchmarks like ChartQA and ChartBench may allow models to leverage pre-existing knowledge, inflating results without truly testing visual capabilities (Yue et al., 2024).

**Proprietary models demonstrate more balanced performance.** Proprietary multimodal models, including the cost-effective GPT-4o mini, achieve competitive results on both traditional chart-related benchmarks and reasoning-intensive tasks like REACHQA and CharXiv. In contrast, open-source models, whether chart-augmented or general-purpose, excel in recognition tasks with simpler charts but struggle in complex ones. This disparity highlights their imbalanced capabilities, and also suggests potential overfitting to traditional benchmarks. Although proprietary models may not always lead in specific tasks, their stable performance across recognition and reasoning tasks indicates better generalization, making them more suitable for real-world applications.

**Specialized training data significantly improves model performance.** Models trained on 8k REACHQA recognition data outperform in recognition tasks, while those trained on 12k reasoning data could do better in reasoning tasks. When both data types are combined (i.e., 20k in total), models see the greatest improvement, with performance increasing by at least 15% across all models we test. Notably, the LLaVA-Next-Llama3-8B model achieves a 34.8% boost in average performance. This suggests that a model's visual capability comprises two complementary aspects, and training on both data types together produces optimal results. Moreover, despite the absence of math-target data in the training set, the models generalize well to the MathVista and MATH-Vision benchmarks, highlighting the transferability of multimodal reasoning abilities distilled from expert trajectories.

## 5 DISCUSSION

### 5.1 DISTILLING EXPERT TRAJECTORIES IMPROVES REASONING

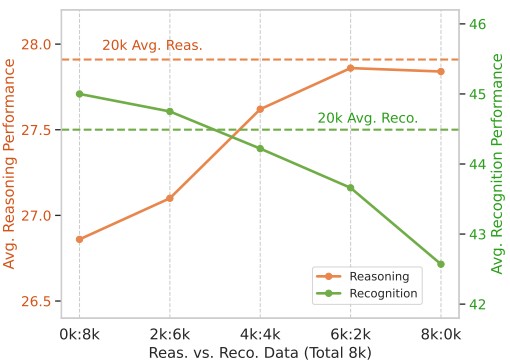

Figure 3: Performance comparison of different Reco.- and Reas.-oriented training data ratios with 8k total data. The dashed line represents the performance with 20k training data in Table 4.

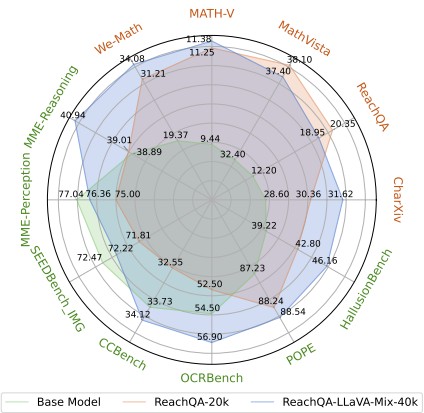

Figure 4: Performance comparison of models on 7 general tasks and 5 reasoning tasks.

To explore the origins of general reasoning abilities, we conduct an experiment comparing several open-source training datasets, including ChartBench (Xu et al., 2023), ChartAst (Meng et al., 2024), Chart-Gemma (Masry et al., 2024b), and The Cauldron (Laurençon et al., 2024). To ensure fairness, we uniformly sample 20,000 instructions from Q&A tasks in each dataset and train LLaVA-Next-Llama3-8B under the same settings. Unlike other individual datasets, The Cauldron is selected for its generality as a collection of 50 vision-language datasets. From

Table 5: Performance comparison of models trained on different datasets. The REACHQA and CharXiv scores refer to Reas. splits here.

| Models | Avg. | REACHQA | CharXiv | MathVista | Math-V |
|---|---|---|---|---|---|
| Base Model | 16.39 | 6.50 | 17.20 | 32.4 | 9.44 |
| + ChartBench | 17.06 | 7.30 | 17.00 | 33.6 | 10.33 |
| + ChartAst | 17.67 | 7.10 | 20.40 | 32.1 | 11.08 |
| + The Cauldron | 18.61 | 10.10 | 19.10 | 35.6 | 9.64 |
| + ChartGemma | 19.11 | 10.00 | 19.40 | 36.4 | 10.62 |
| + REACHQA | **20.74** | **11.10** | **22.50** | **38.1** | **11.25** |

this source, we sample instructions from a mixture of seven chart-related subsets. As shown in Table 5, the model trained on ChartBench performs the worst. This may be due to its instruction responses, which contain only final answers without the reasoning process, limiting the model to learning natural thinking patterns. Although ChartAst includes rationale annotations, their effectiveness is constrained by the simplicity of its template-based questions and a lack of reasoning depth. The model trained on the mixed dataset in The Cauldron shows slightly better performance but is still constrained by the subsets' quality, characterized by simple charts and inadequate rationale annotations. In contrast, the models trained on ChartGemma and REACHQA achieve better performance. This improvement can be attributed to the distillation of high-quality expert trajectories (i.e., from Gemini Flash 1.5 and GPT-4o), which significantly enhance the model's visual reasoning abilities.

While we only use SFT to train the model, incorporating advanced methods like RL (Xi et al., 2024) or DPO (Mitra et al., 2024) could further improve the results. Additionally, the visual richness of charts, as detailed in Appendix G, may enhance the model's generalization abilities.

## 5.2 INTERACTION BETWEEN RECOGNITION AND REASONING ABILITIES

As mentioned before, the recognition and reasoning abilities are likely interdependent. Wang et al. (2024c) also suggest that recognition skills serve as prerequisites for effective reasoning. To explore this further, we conduct an experiment by fixing the total training data at 8k and varying the ratio of reasoning to recognition data from $8:0$ to $0:8$. We train the LLaVA-Next-Llama3-8B model and evaluate them on both recognition-focused benchmarks (i.e., ChartQA, ChartBench, ChartX) and reasoning-focused ones (i.e., REACHQA-Reas., CharXiv-Reas., MathVista).

As shown in Figure 3, increasing the proportion of recognition or reasoning data results in improvements in corresponding task performance. Notably, models trained with a higher ratio of recognition data outperform those trained on 20k total data in recognition tasks. In contrast, as the ratio of reasoning data increases, performance gains plateau and even decline when the ratio reaches 100%.

This suggests that simply increasing the proportion or scale of reasoning-oriented data may lead to diminishing returns. We hypothesize that this is because reasoning abilities are partially dependent on recognition skills. When a model fails to accurately interpret an image, its reasoning process is likely compromised (Wang et al., 2024c). Due to resource limitations, we are unable to scale the total data size further in this study. However, we expect that with larger datasets, the interaction between recognition and reasoning data would become even more significant.

### 5.3 BALANCING GENERAL VISUAL UNDERSTANDING AND REASONING ABILITIES

We are also interested in how models trained on specialized data perform in non-reasoning general-purpose multimodal tasks. To investigate this, we conduct an experiment using 7 popular general multimodal benchmarks—MME-Reasoning, MME-Perception (Fu et al., 2023), SeedBench (Li et al., 2023a), CCBench (Liu et al., 2023), POPE (Li et al., 2023b), HallusionBench (Guan et al., 2024), and OCRBench (Liu et al., 2024c)—along with 5 reasoning-focused benchmarks: REACHQA, CharXiv, MathVista, MATH-Vision, and We-Math (Qiao et al., 2024). And we test three models: the vanilla LLaVA-Next-Llama3-8B, one fine-tuned with 20k REACHQA samples, and another trained on 20k REACHQA samples combined with 20k general-purpose multimodal data randomly selected from the 779k LLaVA-NeXT-Data[2]. This dataset is chosen because the LLaVA-NeXT family of models was officially fine-tuned on it, allowing us to approximate its original data distribution (Li et al., 2024).

The results are shown in Figure 4. We can observe that the vanilla model (green area) struggles with reasoning-intensive tasks, while the model trained on 20k REACHQA data (orange area) improves in reasoning but loses general visual performance. Surprisingly, by incorporating just 20k of general instruction data (blue area), the model not only recovers its general multimodal performance but also retains the enhanced reasoning ability. This results in a well-balanced model with a significant boost in reasoning improvements and minimal drops in general domains.

## 6 CONCLUSION

This paper investigates the challenges complex chart question-answering (CQA) tasks pose for multimodal large language models (MLLMs), focusing on deficiencies in two core abilities: recognition and reasoning. Existing datasets for training these skills often face high annotation costs and an over-reliance on chart data tables, which compromises data quality. To address this, we develop a cost-effective, scalable data synthesis method that distills LLMs' abilities to boost MLLMs' performance. Specifically, we introduce Code-as-Intermediary Translation (CIT), a novel technique that aligns visual and textual modalities through code. Using CIT and LLMs, we create REACHQA, a high-quality multimodal instruction dataset to train and evaluate models. Extensive experiments validated the effectiveness of our method and dataset, demonstrating significant improvements in the performance of the fine-tuned models.

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

## A  CHART TYPES AND TOPICS

We predefined several chart types and topics for Self-Instruct prompting. Table 6 presents the 9 major categories we established, along with their corresponding subcategories. Additionally, Table 7 lists the 38 topics we specified. It is important to note that these topics do not reflect the actual topic distributions in the generated charts, as we encourage the model to combine and expand upon them.

## B  COST OF REACHQA TRAINING DATA CONSTRUCTION

Table 8 provides a detailed expense breakdown. We executed Self-Instruct and Evol-Instruct 3,000 times each to synthesize chart-plotting code, theoretically generating 6,000 charts. However, after accounting for non-executable code and images filtered out by MLLM rating, we ultimately produced 3,249 charts for Q&A synthesis.

Table 6: Major categories and minor categories of charts in REACHQA.

| Major Category | Minor Category |
|---|---|
| Line Charts | line chart, line chart with data annotation, line chart with error bar |
| Pie Charts | pie chart, donut pie chart, sector pie chart, ring chart |
| Bar Charts | bar chart, bar chart with data annotation, stacked bar chart, percentage bar chart, horizontal bar chart |
| 3D Bar Charts | 3D bar chart, stacked 3D bar chart, percentage 3D bar chart |
| Node Charts | directed node chart, undirected node chart |
| Radar Charts | radar chart, radar chart with area filling |
| Area Charts | area chart, stacked area chart |
| Box Charts | vertical box chart, horizontal box chart |
| Scatter Charts | scatter chart, scatter chart with smooth fitting, 3D scatter chart (bubble chart) |
| Specific Charts | heat map, rose chart, funnel chart, waterfall chart, histogram, tree map |

Table 7: Predefined chart topics in Self-Instruct prompting.

| | | |
|---|---|---|
| Art and Design | Futurism and Innovation | Agriculture and Food Production |
| Music and Performance | Astronomy and Space | Transportation and Logistics |
| Business and Finance | Social Media and the Web | Real Estate and Housing Market |
| Travel and Exploration | Society and Community | Government and Public Policy |
| Books and Publishing | Physics and Chemistry | Education and Academics |
| Literature and Writing | Energy and Utilities | Environment and Sustainability |
| History and Culture | Biology and Life Sciences | Language and Communication |
| Architecture and Building | Retail and E-commerce | Social Sciences and Humanities |
| Fashion and Style | Religion and Spirituality | Manufacturing and Production |
| Marketing and Advertising | Food and Beverage Industry | Artificial Intelligence and Robotics |
| Law and Legal Affairs | Healthcare and Health | Human Resources and Employee Management |
| Film and Cinema | Sports and Entertainment | Computer Science and Information Technology |
| Mathematics and Statistics | Science and Engineering | |

## C  ERROR ANALYSIS

To validate the effectiveness of our method and gain deeper insights into model improvements, we conduct a comprehensive error analysis on model predictions before and after REACHQA train-ing. We annotate errors from the test set responses and categorize them into two primary types: recognition errors and reasoning errors using GPT-4o. The results are presented in Table 9.

The results demonstrate that our training yields consistent improvements across different models. For all evaluated models, both recognition and reasoning errors show noticeable reductions after REACHQA training, indicating enhanced capabilities in both aspects. Additionally, stronger base models (e.g., InternVL2-8B) exhibit a lower proportion of recognition errors initially, and after train-ing, the reduction in recognition errors becomes more significant. This suggests that recognition, as a more fundamental capability, could benefit more directly from the training process. These findings align with our observations in § 4.2 and § 5.2.

## D  QUESTION TYPE STATISTICS

To demonstrate the diversity of our synthetic dataset, we conduct an analysis of question types in REACHQA test set. We employ GPT-4o to categorize 2,000 synthetic questions from the test set, equally distributed between recognition and reasoning splits. Table 10 presents the detailed distribution of question types.

## E  DATA CONTAMINATION ANALYSIS

To ensure the validity of our experimental results and exclude potential data contamination, we con-duct a comprehensive analysis of data overlap from both dataset-level and split-level perspectives. First, to evaluate image-level similarity, we employed the SigLIP-400M encoder (Zhai et al., 2023)

Table 8: The average number of input and output tokens is calculated for each step in the REACHQA construction process. In the equation, each term represents the average number of tokens per step (used only in a multi-step framework), while each multiplier corresponds to the number of times that step is executed. The pricing for GPT-4o-2024-08-06 is \$2.50 per 1M input tokens and \$10.00 per 1M output tokens. As a result, the total cost amounts to approximately \$303.95.

| Step | Avg. #tokens of Input | Avg. #tokens of Output | Times | Cost (\$) |
|------|----------------------|------------------------|-------|-----------|
| Self-Instruct | $1,500 + 2,000 = 3,500$ | $500 + 500 = 1,000$ | 3,000 | $\sim 56.25$ |
| Evol-Instruct | $700 + 1,300 = 2,000$ | $300 + 700 = 1,000$ | 3,000 | $\sim 45.00$ |
| Self-Repair | $500$ | $500$ | 1,500 | $\sim 9.38$ |
| Reas-QA-Gen. | $1,000 + 1,500 \times 4 = 7,000$ | $500 + 300 \times 4 = 1,700$ | 3,249 | $\sim 112.09$ |
| Reco-QA-Gen. | $800 + 1,200 \times 4 = 5,600$ | $300 + 200 \times 4 = 1,100$ | 3,249 | $\sim 81.23$ |

Table 9: Error types and proportions in models before and after training with REACHQA.

| Model | # Errors | # Reco. Error | Reco. Proportion | # Reas. Error | # Reas. Proportion |
|-------|----------|---------------|------------------|---------------|--------------------|
| LLaVA-Next-Llama3-8B | 1355 | 827 | 61.0% | 493 | 36.4% |
| LLaVA-Next-Llama3-8B + REACHQA | 886 (-469) | 534 (-293) | 60.3% ($\downarrow$) | 334 (-159) | 37.7% ($\uparrow$) |
| MiniCPM-V2.5-Llama3 | 827 | 513 | 62.0% | 297 | 35.9% |
| MiniCPM-V2.5-Llama3 + REACHQA | 714 (-113) | 414 (-99) | 58.0% ($\downarrow$) | 278 (-19) | 38.9% ($\uparrow$) |
| InternVL2-8B | 655 | 335 | 51.1% | 296 | 45.2% |
| InternVL2-8B + REACHQA | 439 (-216) | 209 (-126) | 47.6% ($\downarrow$) | 198 (-98) | 45.1% ($\downarrow$) |

to generate embeddings for all chart images across different datasets. These embeddings were then projected into a two-dimensional space using t-SNE (Van der Maaten & Hinton, 2008) for visualization, following Xu et al. (2023). Second, we analyzed query-level similarity using the NV-Embed-v2 model (Lee et al., 2024) to generate embeddings for all queries, also visualized through t-SNE.

As shown in Figure 5(a) and (c), the visualization results demonstrate clear distributional differences between REACHQA and existing chart-related benchmarks. While some degree of overlap exists due to the shared nature of chart-related tasks, these instances are limited and do not compromise the overall distinctiveness of our dataset. The distinct clustering patterns in both image and query spaces support the validity of our cross-dataset evaluations and confirm that REACHQA presents novel challenges not fully captured by existing benchmarks.

To address potential data leakage between training and testing splits, which were synthesized through the same process, we conduct a more rigorous analysis as visualized in Figure 5(b) and (d). Beyond the visualization, we compute pairwise similarities between all training and testing samples using the chart embeddings. Among the identified top 50 image pairs with similarity scores exceeding 0.9, our careful manual review revealed only 2 cases with notable similarities. We will exclude them from the test set in future versions and update the evaluation accordingly. For the remaining samples, our review confirmed clear differences in chart topics, data values, and query types, ensuring that no further data leakage or contamination is present.

# F  QUALITATIVE ANALYSIS

To explore the mechanism behind the improved performance of our fine-tuned model, we conduct an analysis of the attention patterns during the next token prediction (Liang et al., 2022; Faysse et al., 2024). Figure 6 presents a comparative case study between the vanilla model and the fine-tuned model. Here, we apply full-parameter fine-tuning instead of LoRA to induce more pronounced changes in the attention layers (Hu et al., 2022). The results show that the vanilla model produces lengthy outputs with redundant analysis and dispersed attention across the image, reaching a wrong conclusion at the end. In contrast, the fine-tuned model identifies the key information at each step, with attention that accurately focuses on relevant visual elements (i.e., labels, axes and values).

This suggests that the model not only imitates expert rationales but also learns the underlying attention patterns crucial for effective visual reasoning. The model automatically establishes a synergistic

Table 10: Question type statistics of REACHQA test set.

| Question Type | # in Reco. split | Question Type | # in Reas. split |
|---|---|---|---|
| Direct Lookup | 177 | Trend Inference | 147 |
| Comparative Lookup | 326 | Causal Analysis | 63 |
| Attribute Identification | 123 | Conditional Reasoning | 55 |
| Pattern Recognition | 61 | Comparative Analysis | 267 |
| Compositionality | 313 | Quantitative Evaluation | 378 |
| Others | 0 | Others | 90 |
| Total | 1,000 | Total | 1,000 |

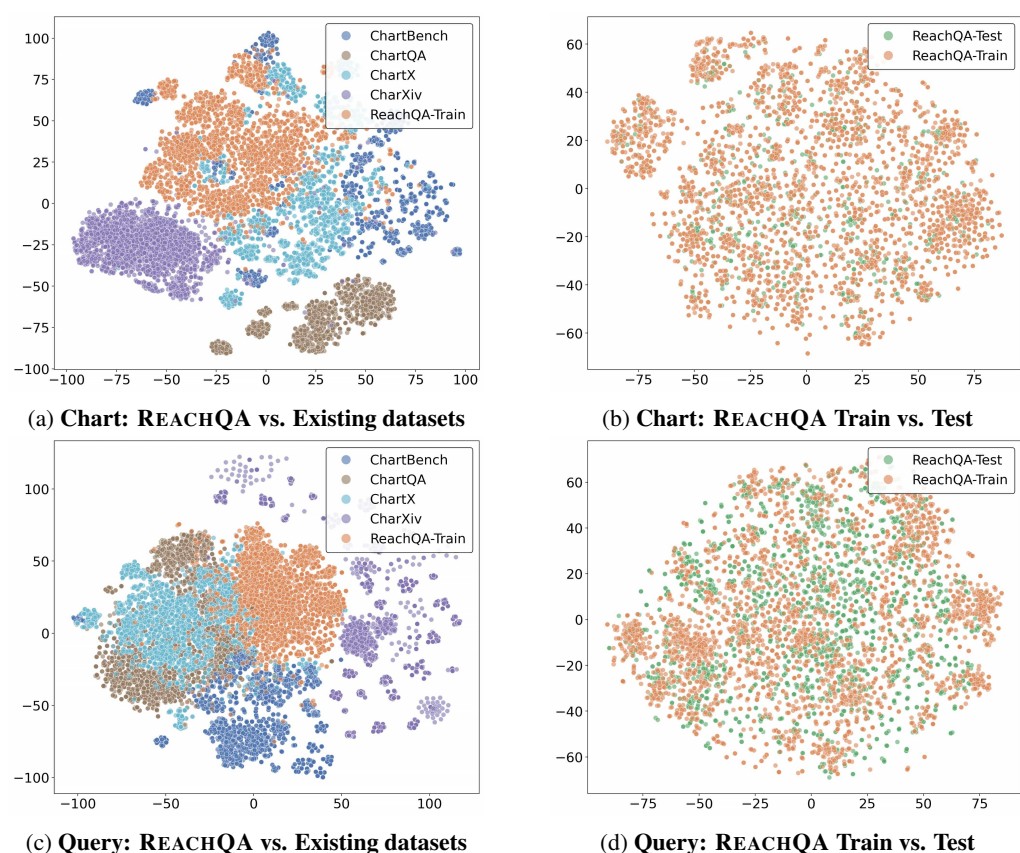

(a) **Chart: REACHQA vs. Existing datasets**

(b) **Chart: REACHQA Train vs. Test**

(c) **Query: REACHQA vs. Existing datasets**

(d) **Query: REACHQA Train vs. Test**

Figure 5: **Data overlap analysis visualization using t-SNE.** We analyze both image-level and query-level similarities through embedding space visualization. (a) and (c) demonstrate the distributional differences between REACHQA and existing datasets, while (b) and (d) examine potential overlap between training and testing splits. The results show clear dataset distinctiveness while revealing expected overlaps due to the shared domain of chart understanding.

relationship between recognition and reasoning capabilities, understanding what to recognize during the reasoning process and utilizing these recognition results to guide subsequent reasoning steps.

## G VISUALIZATION OF CHARTS IN DIFFERENT DATASET

We randomly sample several charts from the training set of ChartQA (Masry et al., 2022), ChartBench (Xu et al., 2023), ChartAst (Meng et al., 2024), ChartGemma (Masry et al., 2024b), and REACHQA. The visualization of the results is presented in Figure 7.

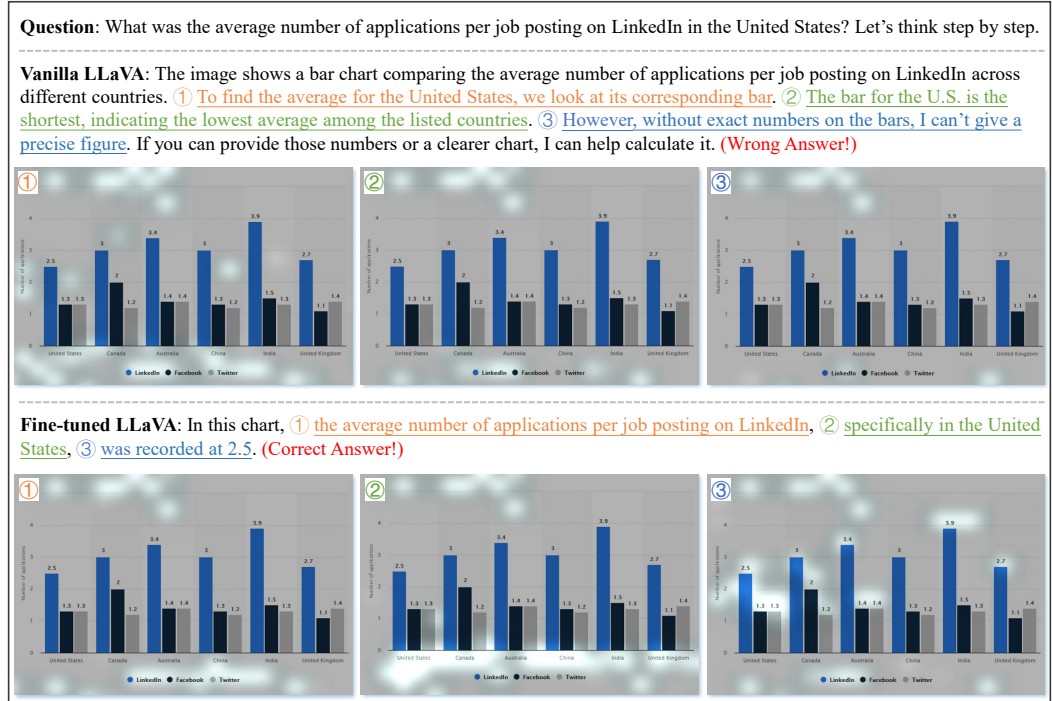

Figure 6: An example of **attention visualization** from the ChartQA dataset. The top row shows the results from the vanilla LLaVA-Next-Llama3-8B model, while the bottom row displays the results from our fine-tuned model. For each output, we present the attention distribution (highlighted zones) at **three key steps**, calculated by averaging the attention values of all tokens in each step.

## H  PROMPT TEMPLATES

Here, we present the prompt templates used in this paper.

### H.1  INTERMEDIARY CODE SYNTHESIS

The prompts used for code generation via the Self-Instruct method are presented in Figure 8, and Figure 9 shows the prompts for the Evol-Instruct method. As illustrated in Figure 10, we utilize four predefined directions to evolve the simple chart-plotting code.

### H.2  BI-DIRECTIONAL TRANSLATION

The prompt used for the Self-Repair method is presented in Figure 11. Additionally, the prompt templates for generating reasoning-oriented questions and answers are listed in Figure 12 and Figure 13. The prompt details for generating recognition-oriented questions and answers are listed in Figure 14 and Figure 15.

### H.3  QUALITY ASSURANCE

The prompt details for rating charts and Q&A are illustrated in Figure 16 and 17.

### H.4  EVALUATION

In the evaluation process, we utilize the LLM-as-a-judge method. The detailed prompt template is illustrated in Figure 18.

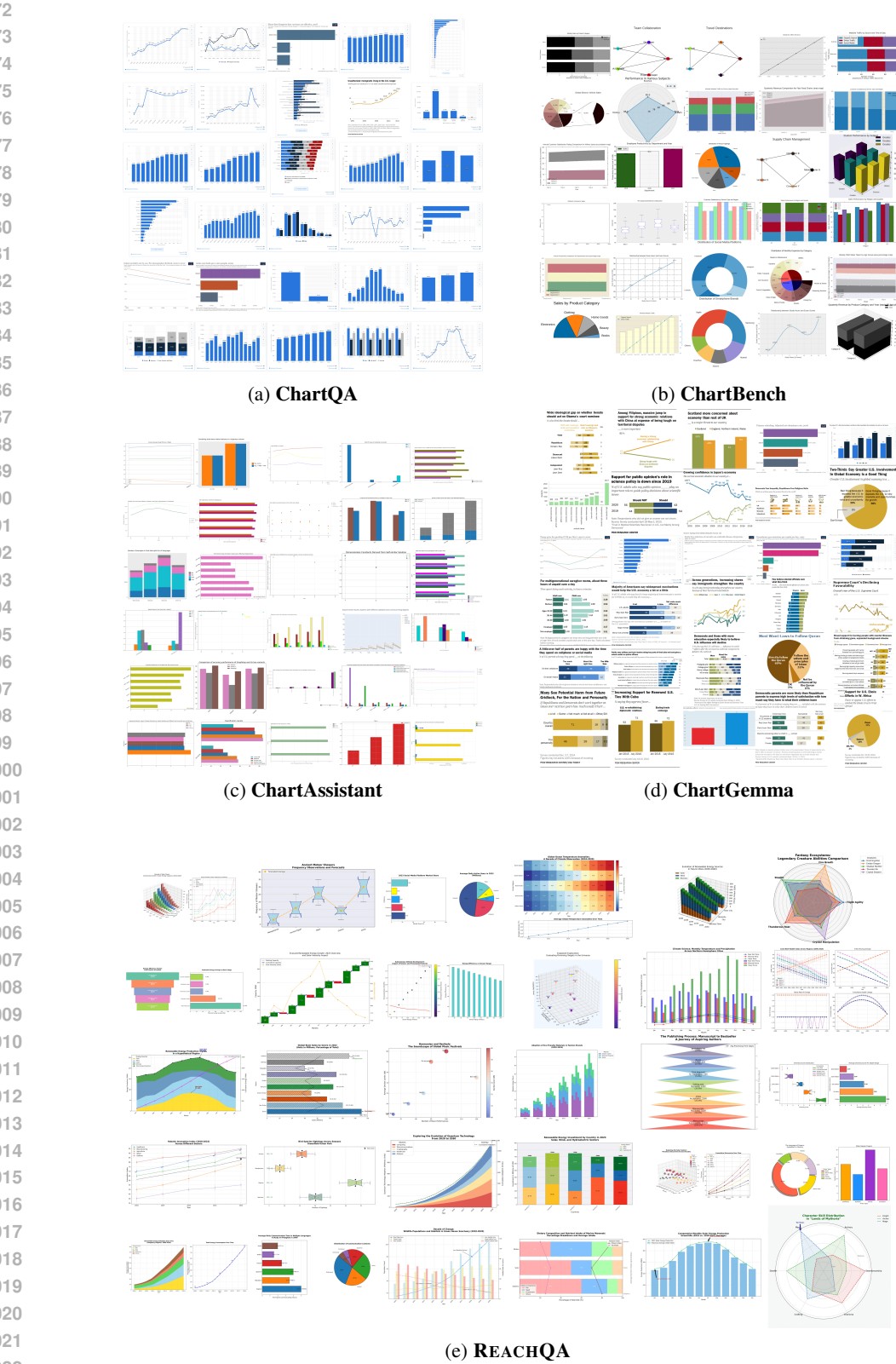

(a) **ChartQA**

(b) **ChartBench**

(c) **ChartAssistant**

(d) **ChartGemma**

(e) **REACHQA**

Figure 7: **Visualizations** of different chart-related training datasets. As shown, REACHQA and ChartGemma exhibit higher chart richness compared to several other datasets. But ChartGemma requires manual collection from multiple sources.

**User:**

As a MatplotLib expert, you are asked to write a new Python plotting script. This script will be used to generate a type-specific chart with artificial data. Here are the requirements:
1. There are several script examples from which you can draw inspiration, but try not to repeat patterns already shown in the examples to maximize diversity.
2. Use the Matplotlib library in Python for plotting. You can use auxiliary libraries such as Numpy, but make sure the code works!
3. The type of chart you need to plot is {type}. Therefore, everything you create must be adapted to fit this type of chart.
4. The topic of the chart can be anything you like, for example, {topic1}, {topic2}, etc.
5. Based on the given chart type and the topic you choose, you need to construct a suitable backstory, which should be reflected in the title, labels, legend, etc.
6. Based on the backstory, you need to construct contextual data inputs in the form of Python lists or Numpy arrays. Information contained in the data can be adapted as appropriate to fit the type of chart.
7. You must not use random() to construct the data, as it needs to be explicitly created regardless of your chart type and topic.
8. Be as imaginative and creative as possible in drawing the chart, both in terms of data and plotting details.

Here are three examples to consider:
{demo1}
{demo2}
{demo3}

Now, let's take this task step by step. First, we have to plan out the title and backstory of the chart and create data based on the above.

**Assistant:**

{model_response}

**User:**

Please complete the entire script by plotting a chart based on the data generated. Here are some highlighted requirements and notes.

Requirements:
1. If you find that the generated data is not appropriate while plotting the chart, modify it further as needed.
2. The information on the chart should be complete enough to be understandable, but avoid including the full backstory or too much text in the figure.
3. Avoid occlusion of visual elements. If necessary, automatically adjust the image layout before plt.show() using tight_layout().
4. If the text in the chart is too long, find a way to make it all visible instead of overlapping. If the title is too long, you can break it into multiple lines.
5. Once again, be as imaginative and creative as possible in creating the details of the chart.
6. Above all, double-check to ensure the code works. Reduce unnecessary comments and focus on functionality.

Now, generate your final plotting script in a single python code block.

Figure 8: Prompt template for code generation via Self-Instruct method.

**User:**

As a MatplotLib expert, you are asked to optimize a Python plotting script to make the plotted chart more complex. The script will be used to generate charts for a mathematical test, so you should make it a little more challenging.

This is the code you need to optimize:
`{code}`

Here's what I'd like you to do to optimize the chart: `{direction}`

Now, let's take this task step by step. First, please read the given code carefully and analyze the chart it draws. Then, think about your optimization ideas with the given directions.
In this step, you don't need to give the final code, only show the design ideas.

**Assistant:**

`{model_response}`

**User:**

Please implement the final optimized script based on the above design ideas combined with the original code.

Remember:
1. Avoid visual elements that obscure each other, e.g., legends, labels. Automatically adjust the image layout before plt.show() using tight_layout(). if necessary.
2. If the text in the chart is too long, find a way to make all the text show up instead of overlapping. If the title is too long, you can break it into multiple lines.
3. Be as imaginative and creative as possible in creating details of the chart, but don't make the chart redundant just to cope.
4. If you are adding a new plot, take care that the chart is complete with all the elements, such as labels, axes, legends, and colors, unless it is intended to be shared with the original chart.
5. If you are adding a new plot, carefully construct meaningful data and consider whether to give the new sub-plot a sub-title.
6. You must not use random() to construct the data, as it needs to be explicitly constructed regardless of your chart type and topic.
7. Above all, double-check to make sure the code works. Reduce unnecessary comments and focus on functionality.

Now, generate your optimized plotting script in a single python code block.

Figure 9: Prompt template for code generation via Evol-Instruct method.

**Evolution Direction:**

- Increase the size of the input data or the number of data groups as appropriate so that it requires a higher level of mathematical understanding. Note if there is a sum requirement.
- Try changing or adding some visual elements to make visual effect better. The elements you add must make sense and not be redundant.
- Incorporate an overlay plot of a different type on the original chart. Use related but not identical data for the added plot.
- Extend an additional subplot of a different type beside the original chart (2 in total). Use related but not identical data for the added plot.

Figure 10: Predefined evolution directions for Evol-Instruct method.

**User:**

As a Python and Matplotlib expert, you have been asked to fix the following code. The error code is:
`{code}`

The code reports the following error message when run: `{error}`

Please analyze the error first, and then provide the revised code within a single Python code block. There should only be one Python code block in your response, containing the complete revised code.

Figure 11: Prompt template for Self-Repair.

**User:**

You are both an expert Matplotlib plotter and a professional maths teacher. Now, you are asked to generate a mathematical reasoning question about a given chart. This chart and question will be used as a question on this year's college admissions examination. As a question writer, you need to ensure that the question is challenging yet fair, testing the students' ability to analyze data, interpret trends, and apply mathematical concepts.

First, please read the following plotting script in Python, try to visualize the figure in your mind and to understand the meaning of the chart. After you've analyzed this chart, we'll start generating the associated question.

Here are some tips for you:
1. The plotting script (including the code itself, data mapping and labels) is absolutely correct and you can trust it completely.
2. The question need to be based on the chart type, chart topic, and the given data. It can relate to the chart as a whole or to localized details, so you need to look closely.
3. The question should be challenging, requiring visual observation skills and mathematical reasoning skills. So you need to have an deep understanding of the chart.
4. If there is no data annotation in the figure, try not to generate questions that require too many numerical recognition to reduce inconsistent answers due to visual errors.
5. If some numerical recognition is needed, choose distinguishable colors, lines, heights, and other features that make it easy to estimate without data annotation.
6. You don't need to describe the content of the figure in the question text. This can be left for students to think about.

Here is the plotting script:
{code}

Now, please generate 4 questions at a time, each of which needs to look at a different aspect of the chart, in the following format:
Question 1:
Question 2:
Question 3:
Question 4:

Figure 12: Prompt template for generating reasoning-oriented questions.

1242
1243
1244
1245
1246
1247
1248
1249
1250
1251
1252
1253
1254
1255
1256
1257
1258
1259
1260
1261
1262
1263
1264
1265
1266
1267
1268
1269
1270
1271
1272
1273
1274
1275
1276
1277
1278
1279
1280
1281
1282
1283
1284
1285
1286
1287
1288
1289
1290
1291
1292
1293
1294
1295

**User:**

You are both a Matplotlib graphing expert and a professional math teacher. Now, you have been asked to generate an answer to a given chart and question. This chart and question will be used as a question on this year's college admissions examination. As the answer writer, you need to ensure that the answer is both correct and detailed, and educational.

First, please read the following plotting script in Python, try to visualize the figure in your mind and to understand the meaning of the chart. After you've analyzed this chart, we'll start generating the answer.

Here is the plotting script:
{code}

Here are some tips for you to generate the answer:
1. First and foremost, the answer needs to be based on the chart information.
2. In the answer, you will also need to solve the question step-by-step, including reasoning steps and recognition steps (but keep concise).
3. You need to explicitly involve a final answer; the type of answer can be a certain number, a noun, or Yes/No, etc.
4. The answer should contain multiple reasoning or calculation steps and be presented in an understandable and educational paragraph.
5. NEVER include any information relating to the Python script in the question or answer, as students will ONLY have access to the plotted figure.

Here is the question: {question}

Now, you can start to generate the answer. Your output needs to follow this format:
Answer:

Figure 13: Prompt template for generating reasoning-oriented answers.

**User:**

You are both an expert Matplotlib plotter and a professional maths teacher. Now, you are asked to generate a recognition-oriented question about a given chart. This chart and question will be used as a question on this year's elementary math examination to test students' ability to read charts.

First, please read the following plotting script in Python, try to visualize the figure in your mind and to understand the meaning of the chart. After you've analyzed this chart, we'll start generating the associated question.

Here are some tips for you:
1. The plotting script (including the code itself, data mapping, and labels) is absolutely correct and you can trust it completely.
2. Descriptive questions are questions that can be answered based on basic chart information, such as titles, labels, tick marks, colors, etc.
3. The generated Q&A needs to be based on the chart type and data. It should be answerable through visual observation.
4. If there is no data annotation in the figure, try not to generate questions that require too many numerical recognitions to reduce inconsistent answers due to visual errors.
5. If some numerical recognition is needed, choose distinguishable colors, lines, heights, and other features that make it easy to estimate without data annotation.
6. You don't need to describe the content of the figure in the question text. This can be left for students to think about.
7. This question needs to explicitly involve a final answer; the type of answer can be a certain number, a noun, or Yes/No, etc.
8. NEVER include any information relating to the Python script in the question or answer, as students will ONLY have access to the plotted figure.

Here are some examples of descriptive questions:
- How many colors are used in the chart? How many city categories are in the chart?
- What's the leftmost value of the bar in China? And what is the value of the bar next to it?
- For the subplot at row 2 and column 1, what is the minimum value of the solid line?
- Which name does the second largest sector represent? What is its value?
- Does the blue triangle in the chart represent a higher value than the red circle?

Here is the plotting script:
{code}

Now, please generate 4 questions at a time, each of which needs to look at a different aspect of the chart, in the following format:
Question 1:
Question 2:
Question 3:
Question 4:

Figure 14: Prompt template for generating recognition-oriented questions.

**User:**

You are both a Matplotlib graphing expert and a professional math teacher. Now, you have been asked to generate an answer to a given chart and question. This chart and question will be used as a question on this year's elementary math examination to test students' ability to read charts. As the answer writer, you need to ensure that the answer is both correct and detailed, and educational.

First, please read the following plotting script in Python, try to visualize the figure in your mind and to understand the meaning of the chart. After you've analyzed this chart, we'll start generating the answer.

Here is the plotting script:
`{code}`

Here are some tips for you to generate the answer:
1. First and foremost, the answer needs to be based on the chart information.
2. In the answer, you will also need to solve the question step-by-step, including reasoning steps and recognition steps (but keep concise).
3. You need to explicitly involve a final answer; the type of answer can be a certain number, a noun, or Yes/No, etc.
4. The answer should contain multiple reasoning or calculation steps and be presented in an understandable and educational paragraph.
5. NEVER include any information relating to the Python script in the question or answer, as students will ONLY have access to the plotted figure.

Here is the question: `{question}`

Now, you can start to generate the answer. Your output needs to follow this format:
Answer:

Figure 15: Prompt template for generating recognition-oriented answers.

**User:**

`<image>`

You are a strict MatplotLib plotter and have been asked to evaluate the given chart. Rate the chart from 1 to 5 based on these criteria:

**1 point**: This chart is the poorest in quality and fails to accurately represent any relevant data. It is characterized by a complete breakdown in visual representation; elements are cluttered, text heavily overlaps, legend is missing, or large areas are left blank, making the chart unreadable. The design shows no understanding of effective data visualization practices.

**2 points**: The chart displays incorrect or irrelevant visual elements, with significant inaccuracies that misrepresent the data. The layout suffers from clutter, substantial overlapping of text and other visual elements, such as the legend or labels, and poorly designed axes that result in uneven distribution, severely impeding accurate interpretation.

**3 points**: This chart represents some correct data points but makes basic errors in visual representation. It may use misleading scales, inappropriate chart types, omit key data. Visual clutter and overlapping elements, such as text obscuring parts of the chart or sub-diagrams overlapping each other, detract from the chart's clarity and readability.

**4 points**: The chart accurately represents most of the major data points and important details of the dataset. Minor visual errors exist, such as slight occlusions of text or sub-optimal positioning of elements like legends or labels, but these do not significantly affect the overall accuracy or readability. The chart demonstrates a good understanding of effective visualization techniques but could still be improved in terms of visual layout and the balance of details.

**5 points**: This is an exemplary chart that perfectly encapsulates all critical data points and relationships with outstanding visual clarity and no occlusions. It demonstrates a thorough understanding of data visualization techniques, making excellent use of space and visual elements. The chart is informative, clear, engaging, and free from any visual errors.

Score the chart on this scale, providing a short analysis and a single value. Your response should be in the format:
Analysis: (your analysis)
Rating: (int)

Figure 16: Prompt template for rating the chart quality.

> **User:**
>
> ```
> <image>
> ```
>
> You are a visual question answering (VQA) data annotator. Your task is to review the following chart and question, and determine if the answer is correct based on the information in the chart. You should carefully analyze the chart, taking into account all relevant data points, labels, and trends. Then, conduct an in-depth analysis to determine if there are any unreasonable or incorrect aspects in the figure, question, or answer.
>
> Specifically, consider the following points:
> 1. Are the provided question and answer relevant to the chart? Can the answer be found in the chart?
> 2. Do the colors in the charts and questions correspond correctly? Are there instances where the colors are incorrectly referred to?
> 3. Do the data in the charts and questions correspond correctly? Are there any errors in the data or misalignment of information?
> 4. Is the provided answer correct? Are there any logical errors or unreasonable points?
> 5. Apart from the points listed above, is there anything else in this question and answer that doesn't make sense?
>
> Here is the question and answer about the given chart:
> Question: {question}
> Answer: {answer}
>
> You are asked to provide a short analysis and decide whether to keep the example. Your response should be in the format:
> Analysis: (your analysis)
> Decision: (yes/no)

Figure 17: Prompt template for rating Q&A quality.

> **User:**
>
> Compare the ground truth with the prediction from AI model and determine if the prediction is correct. The question is about an image, which we have not given here. You need to determine whether the model's prediction is consistent with the ground truth. No points will be awarded for wrong answers, over answers or under answers. The reasoning process in the prediction does not need to be considered too much, you only need to determine if the final answer is consistent. There are times when the answer may have a different form of expression and some variation is acceptable.
>
> ## Question: {question}
> ## Ground Truth: {answer}
> ## Prediction: {prediction}
>
> Now, let's analyze it and then provide your judgment. Your response must follow the format below:
> Analysis: (analyze the correctness briefly)
> Correctness: (Yes or No)

Figure 18: Prompt template for evaluating the model prediction with LLMs.

