# OpenReview forum: "Distill Visual Chart Reasoning Ability from LLMs to MLLMs"
_ICLR.cc/2025/Conference — Submitted to ICLR 2025_

### Official Review · Reviewer_Yw4L · 2024-11-02

**Soundness:** 3
**Presentation:** 3
**Contribution:** 3
**Rating:** 8
**Confidence:** 4

**Summary:**

This paper proposes an efficient method (CIT) that aims to use code as a bridge and incorporates certain characteristics of diagram research (such as visual complexity) to generate data (REACHQA). On this basis, the paper deconstructs the capabilities required for the Chart QA task into recognition and reasoning, and demonstrates through experiments that the proposed method is effective.

**Strengths:**

1. CIT incorporates the characteristics of the chart tasks to generate high-quality data.
2. This paper attempts to deconstruct the capability of chart-based question answering into recognition and reasoning and provides thorough justification based on this approach.

**Weaknesses:**

1. The authors should consider and attempt to provide data on the overlap between the generated data, the validation set, and relevant benchmark datasets to further substantiate the validity of the experiments.
2. I hope the authors can provide more evaluation results on benchmarks that are specific to recognition [1] and reasoning [2] to demonstrate that the method has strong generalization capabilities. If the results do not decline, that would be an even more interesting observation.

[1] OCRBench: On the Hidden Mystery of OCR in Large Multimodal Models
[2] We-Math: Does Your Large Multimodal Model Achieve Human-like Mathematical Reasoning?

**Questions:**

Please refer to the weaknesses section.

---

> ### Author Response · Authors · 2024-11-19
> **Official Response from Authors.**
>
> We appreciate your valuable time, insights, and for highlighting our strengths (i.e., **the CIT method** and **deconstruction of visual reasoning ability**). In the following, we will carefully respond to your questions.
>
> ---
>
> 1. **Question about the overlap between existing datasets.**
>
>    - Thank you for your valuable suggestion! We conduct a comprehensive analysis of the data overlap as follows:
>      - **Image similarity**: Following ChartBench [1], we apply the SigLIP-400M encoder [2] to generate embeddings for images across datasets and visualize using t-SNE [3].
>      - **Query similarity**: Similarly, we apply the NV-Embed-v2 model [4] to generate embeddings for the queries across datasets, also visualize using t-SNE.
>      - We compare (1) ReachQA training set vs. other chart-related benchmarks, and (2) ReachQA training set vs. ReachQA test set. **The visualization results have been updated to Appendix E (Page 16) of the revised paper**.
>    - **Key findings**:
>      - Our dataset exhibits clear distinctions from existing chart-related benchmarks in both images and queries. While some degree of overlap is naturally expected due to the shared focus on chart-related tasks, these instances are limited and do not compromise the broader differences in distribution. Overall, we believe that it shows the validity of our main experiments.
>      - Since our training and testing sets were synthesized using the same process—though independently conducted—some distribution overlap is expected. **To address concerns regarding potential data leakage, we implemented a filtering and manual review process**. Specifically, we identified the top 50 most similar image pairs (similarity > 0.9) and carefully examined these pairs along with their associated queries. Among these, only 2 examples exhibited notable similarities. We will exclude them from the test set in future versions and update the evaluation accordingly. For the remaining samples, our review confirmed clear differences in chart topics, data values, and query types, ensuring that no further data leakage or contamination is present.
>
> 2. **Question about the additional benchmark evaluations.**
>
>    - In response to your suggestion, we have added additional benchmark evaluations following the setting of Section 5.3, with results provided as follows:
>
>      | Models                  | OCRBench |  We-Math  |
>      | :---------------------- | :------: | :-------: |
>      | LLaVA-Next-Llama3-8B     |   54.5   |   19.37   |
>      | + ReachQA-20k           |   49.2   |   31.21   |
>      | + ReachQA-LLaVA-Mix-40k | **56.4** | **34.08** |
>
>    - **Consistent with the trends observed in our paper**, we found some performance decline on OCRBench after training on ReachQA. This is likely due to the significant difference in task objectives, as OCRBench focuses on direct recognition. However, performance quickly recovered and even surpassed the base model once fine-tuned with a mixed dataset incorporating general-purpose data. For reasoning-intensive tasks like We-Math, the fine-tuned model achieved better performance at first, and this improvement remained stable after incorporating general data.
>
> ---
>
> ## Reference
>
> [1] ChartBench: A Benchmark for Complex Visual Reasoning in Charts, arXiv:2312.15915
>
> [2] Sigmoid Loss for Language Image Pre-Training,  ICCV 2023
>
> [3] Visualizing Data Using t-SNE, JMLR 2008
>
> [4] NV-Embed: Improved Techniques for Training LLMs as Generalist Embedding Models, arXiv:2405.17428

---

> > ### Comment · Reviewer_Yw4L · 2024-11-26
> >
> > Thank you for your response. It addresses my main concern (w1).
> >
> > Regarding the results related to We-Math and OCRBench mentioned in your response, I find this phenomenon quite interesting. We-Math does not include chart data and focuses on process supervision—breaking down complex problems into the necessary steps for solving them. It shows a significant improvement on ReachQA-20K, which is a dataset based on chart Q&A tasks.
> >
> > **I hope the authors can present more results for both We-Math and OCRBench following the setting in Figure 3, particularly including detailed metrics for IK, IG, CM, and RM, before the deadline.**
> >
> > Overall, I'm satisfied with the authors' response and will raise my score.

---

> ### Author Response · Authors · 2024-11-26
> **Official Response from Authors.**
>
> Thank you for your practical and professional suggestions, which helped us make our paper better and more thoroughly researched! We are also pleased that our previous clarifications were satisfactory. **[Updated]** In the following, we will carefully respond to your follow-up questions.
>
> ---
>
> **Question about detailed results on additional benchmark**.
>
> - In response to your suggestions, we conducted the corresponding experiments on OCRBench and We-Math based on the setting in Figure 3. The results are as follows:
>
>   > **Note:** During the OCRBench experiments, we identified and fixed a bug in [VLMEvalKit](https://github.com/open-compass/VLMEvalKit), the evaluation toolkit used in our study. This bug occasionally caused our fine-tuned model to produce empty string outputs due to formatting. Results marked with (*) reflect the corrected scores, which do not alter the conclusions presented in our previous analysis.
>
>   | Data Size | Reas. vs. Reco. | OCRBench (Acc.) | We-Math (Acc.) | Insufficient Knowledge (↓) | Inadequate Generalization (↓) | Complete Mastery (↑) | Rote Memorization (↓) |
>   | :-------: | :-------------: | :-------------: | :------------: | :------------------------: | :---------------------------: | :------------------: | :-------------------: |
>   |     -     |        -        |      54.5       |     19.37      |        83.05% (436)        |        **3.05% (16)**         |      7.24% (38)      |      47.95% (35)      |
>   |    20k    |    12k : 8k     |      52.5*      |     31.21      |        71.81% (377)        |          7.24% (38)           |     14.10% (74)      |      32.73% (36)      |
>   |    40k    |    12k : 8k     |   **56.9**\*    |   **34.08**    |      **68.00% (357)**      |          8.38% (44)           |     17.14% (90)      |      27.42% (34)      |
>   |    8k     |     0k : 8k     |      54.3       |     33.74      |        69.90% (367)        |          7.24% (38)           |     17.14% (90)      |      25.00% (30)      |
>   |    8k     |     2k : 6k     |      54.4       |     33.28      |        70.29% (369)        |          7.62% (40)           |     16.57% (87)      |      25.00% (29)      |
>   |    8k     |     4k : 4k     |      53.9       |     31.72      |        72.76% (382)        |          6.86% (36)           |     15.05% (79)      |      26.17% (28)      |
>   |    8k     |     6k : 2k     |      53.3       |     32.82      |        70.29% (369)        |          7.24% (38)           |     16.57% (87)      |      26.27% (31)      |
>   |    8k     |     8k : 0k     |      52.4       |     33.28      |        69.33% (364)        |          8.00% (42)           |   **17.33% (91)**    |    **23.53% (28)**    |
>
> - **Key findings**:
>   - The OCRBench (Acc.) results are **consistent with the observations in Figure 3**. The model’s recognition performance strongly correlates with the proportion of recognition-oriented training data. Notably, excessive training or reasoning data ratio may lead to performance drops.
>   - The We-Math (Acc.) results show greater variance but **consistently outperform the vanilla model across all settings**, demonstrating the generalizability of our method despite the domain differences. However, achieving optimal performance requires careful control of data volume and ratio.
>   - On the additional evaluation metrics of We-Math (IK, IG, CM, RM), most results align with the observations of the original paper, such as the **shifted challenges from IK to IG**. Notably, we found that **the model trained with 100% reasoning data demonstrated enhanced output reliability**, with CM (↑) and RM (↓) reaching optimal values. This suggests improved robustness in reasoning processes (e.g. correct intermediate steps leads to correct final answers).
>
> ---
>
> We will continue to refine this work to make it more worthy of acceptance. Thank you again for your consideration!

---

> > ### Comment · Reviewer_Yw4L · 2024-11-28
> >
> > Thank you for your response. It addresses all of my concerns.
> >
> > I find the conclusions drawn from the results of the additional benchmarks on recognition and reasoning to be an interesting phenomenon. I will maintain my current rating and increase my confidence.

---

> > > ### Author Response · Authors · 2024-11-28
> > > **Thank you!**
> > >
> > > Thank you for your kind words and for taking the time to review our response. We are glad that your concerns have been addressed, and we truly appreciate your recognition of our work. We will carefully revise the paper based on your valuable feedback. Thank you again for your consideration!

---

### Official Review · Reviewer_4dEr · 2024-11-03

**Soundness:** 3
**Presentation:** 3
**Contribution:** 3
**Rating:** 6
**Confidence:** 4

**Summary:**

This paper purpose CiT (Code-as-Intermediary Translation), a data synthesis method for distilling visual reaoning abilities from LLM to MLLMs. In CiT, code are rendered to charts and translated to instruction tuning by LLMs. Using this pipeline, this paper developed ReachQA, which is an effective benchmark and training dataset.

**Strengths:**

- The overall writing is clear and easy to follow.
- The idea of CIT is straightforward, effective and easily scalable.
- CIT addressed an important gap in MLLM training in lacking accurate textual annotations of visual diagrams.
- The resulting dataset has high quality. Althought only with 3K images and 20K QAs, the perforamnce of MLLMs improves by at most 35 percent. In addition, when trained a mixture with general multimodal dataset, the model can effectively retain its general MM benchmarks.

**Weaknesses:**

- As authors claimed, the MLLMss ability consist of two main parts, 1. recognizing key information from visual inputs 2. conducting reasoning over it. The design of ReachQA is rich in both parts, so it's unclear which part improves models' over performance the most, the reviewer is aware that similar analysis is conducted in section 5, but an error analysis (similar with figure 1) on before/after ReachQA training can make it more clear.
- Dataset volume is a concern. As a training dataset, only 3K images is not sufficient, especially for cross-modal training. So it's likely that the improvement gained by training on ReachQA is that the model learns reasoning better rather than recogonition.

**Questions:**

Please refer to above.

---

> ### Author Response · Authors · 2024-11-19
> **Official Response from Authors.**
>
> We appreciate your valuable time, insights, and for highlighting our strengths (i.e., **the writing**, **the idea of CIT**, and **the effectiveness of our dataset**). In the following, we will carefully respond to your questions.
>
> ---
>
> 1. **Question about error analysis on models before/after ReachQA training.**
>
>    - Thank you very much for your suggestion! We have added an analysis of errors before and after training to provide a clearer understanding of the improvements. The results are as follows:
>
>      | Model                          |  # Errors  | # Reco. Error | Reco. Proportion | # Reas. Error | Reas. Proportion |
>      | ------------------------------ | :--------: | :-----------: | :--------------: | :-----------: | :--------------: |
>      | LLaVA-Next-Llama3-8B           |    1355    |      827      |      61.0%       |      493      |      36.4%       |
>      | LLaVA-Next-Llama3-8B + ReachQA | 886 (-469) |  534 (-293)   |    60.3% (↓)     |  334 (-159)   |    37.7% (↑)     |
>      | MiniCPM-V2.5-Llama3            |    827     |      513      |      62.0%       |      297      |      35.9%       |
>      | MiniCPM-V2.5-Llama3 + ReachQA  | 714 (-113) |   414 (-99)   |    58.0% (↓)     |   278 (-19)   |    38.9% (↑)     |
>      | InternVL2-8B                   |    655     |      335      |      51.1%       |      296      |      45.2%       |
>      | InternVL2-8B + ReachQA         | 439 (-216) |  209 (-126)   |    47.6% (↓)     |   198 (-98)   |    45.1% (↓)     |
>
>    - The results demonstrate that: (1) For different models, both types of errors show a noticeable reduction, indicating that **the trained model improves in both recognition and reasoning abilities**. This is consistent with the observations in Sections 4.2 and 5.2 of our paper. (2) Additionally, stronger base models (e.g. InternVL2-8B) exhibit a lower proportion of recognition errors, and after training with ReachQA, the reduction in recognition errors is more significant. This suggests that **recognition, as a more fundamental capability, could benefit more directly from the training**.
>
> 2. **Question about dataset sufficiency for cross-modal training.**
>
>    - Thank you for pointing this out! We acknowledge that the ReachQA dataset volume is still limited, especially for training larger MLLMs. However, as our method focuses on automated data synthesis, it is feasible to scale this approach to larger datasets. **The experiments in our paper can be seen as an initial attempt under resource constraints**, and we will release our implementation to support future efforts to expand data volume.
>    - Additionally, based on prior experience with MLLM training, we agree that data size is crucial for enhancing recognition abilities. That said, we also believe that **a visually complex image can be worth many simpler ones**. For instance, each chart in ReachQA is paired with six distinct questions on average, covering focuses from local to global, and information types from colors to numeric values and their combinations. These diverse questions require the model to recognize various aspects of the image, so that each image to be fully utilized during training.
>    - Regarding the performance gained by training on ReachQA, as discussed in Section 5.2 and the supplementary error analysis, these two abilities—recognition and reasoning—**likely interact and mutually reinforce one another**. As the model’s recognition skills improve, more accurate identification can support more effective reasoning; likewise, improved reasoning abilities can help the model better identify the elements it needs to recognize in subsequent steps.

---

> > ### Comment · Reviewer_4dEr · 2024-11-25
> >
> > Thanks for the response. The error analysis can help us better understand the source of the improvement. In addition, it would be good to provide some qualitative examples to give us better intuition about the performance gain.
> >
> > Thanks for the clearification for dataset sufficiency, and I agree that the dataset generated are in good quality given the performance. It seems that the main bottleneck of generating more data is the cost, has the author measure the feasibility of cheaper API models (e.g. GPT-4o-mini) or open-sourced models (e.g. Qwen-2.5)?
> >
> > Overall I'm satisfied with the response and will keep my positive rating.

---

> ### Author Response · Authors · 2024-11-25
> **Official Response from Authors.**
>
> Thank you for your valuable feedback and your positive assessment of our first-round response! We are pleased that our previous clarifications were satisfactory. In the following, we will carefully respond to your follow-up questions.
>
> ---
>
> 1. **Question about qualitative examples**.
>
>    - Thank you very much for your suggestion! To provide intuitive insights into the performance gains, we have conducted **a case study from the perspective of attention**. The study aims to visually demonstrate the mechanisms behind the improved performance of our fine-tuned model. The results have been added to **Appendix F (Page 16) and Figure 6 (Page 18) of the revised paper** for your review.
>
>    - Specifically, we analyzed the model’s attention patterns to the image during next-token prediction [1,2], comparing the vanilla model with our fine-tuned version. Figure 6 presents a case where the vanilla model produces **lengthy, redundant outputs with scattered attention** and ultimately reaches incorrect conclusions. In contrast, the fine-tuned model effectively **focuses its attention on key elements (e.g., labels, axes, values) at each step**, leading to accurate reasoning.
>
>    - This analysis indicates that the fine-tuned model not only mimics expert rationales but also **develops robust attention patterns**, enabling a synergistic interplay between recognition and reasoning. It identifies crucial elements during reasoning and utilizes them to guide subsequent steps.
>
> 2. **Question about the cost and alternative models.**
>    - Thank you for raising this insightful point! Our approach is already cost-effective for reasoning-intensive multimodal data synthesis. Using GPT-4o, we were able to generate 3k images and 20k instructions for approximately $300—**a fraction of the cost compared to manually collecting and annotating complex charts**.
>    - We have also experimented with several open-source models, such as Llama-3.1-70B-Instruct and Qwen-2.5-72B-Instruct. For intermediary code synthesis, these models initially achieved a pass rate of around 60%, which improved to over 90% through iterative self-repair. This demonstrates that some open-source models are **already feasible for further reducing costs**.
>    - However, for instruction generation, these models performed significantly worse than GPT-4o or Claude 3.5 Sonnet. This limitation is likely due to their **weaker understanding of complex code**—particularly the evolved versions after our Evol-Instruct process—making it challenging for them to effectively “visualize” charts in their “mind”.
>
> ---
>
> ## Reference
>
> [1] EViT: Expediting Vision Transformers via Token Reorganizations, ICLR 2022
>
> [2] ColPali: Efficient Document Retrieval with Vision Language Models, arXiv.2407.01449

---

### Official Review · Reviewer_cHNi · 2024-11-04

**Soundness:** 3
**Presentation:** 3
**Contribution:** 3
**Rating:** 5
**Confidence:** 3

**Summary:**

This paper aims to distill the visual chart reasoning ability from LLMs to MLLMs through a code-as-intermediary translation (CIT) method. The motivation behind the work lines the concern of current MLLMs and then distill the LLMs knowledge about reasoning to MLLMs is a way. The method separates two steps, recognition and reasoning, then the authors try to generate more codes for chart generation, and then generate instruction-answer pairs. Besides, the authors also propose a ReachQA dataset for training and evaluation. Through multiple experiments, the authors demonstrate the effectiveness of the method, the dataset with strong performances.

**Strengths:**

1. The inspiration of the method is interesting and leverages the translation principle to incorporate an intermediate language. By using code as an intermediate language, the method is interesting and novel.
2. The steps behind the method are clear and reasonable, first use code to generate charts, and then generate QA pairs. The authors use multiple methods, such as Evol-instruct, self-instruct, llm-as-judge and so on to make the generated data to be high quality and diversity.
3. The experiments are extensive to show the effectiveness of the method and the study is also good.

**Weaknesses:**

1. The main concern is about the motivation or the story. While the authors mention that the existing MLLMs are struggle in the recognition and reasoning abilities, the authors then want to use distill LLMs' reasoning ability to MLLMs. However, it is clear that Claude and GPT-4o are smart in these ways. Therefore, the problem is not about existing MLLMs but the open-sourced, or freely sourced MLLMs. I encourage the authors to rephrase the story, which then could introduce the method more clearly.
2. Though the steps are great and the authors use multiple existing methods to generate and guarantee the data quality. The novelty of the method/framework is clearly not that big, but I like the code-as-intermediate translation method.
3. In detail, the analysis of the question types should be reported, which could show the diversity and quality; from the experiments, strong improvements are achieved but still largely fail behind the close models; the self-repair method is good but the prompt is too simple, I am curious about other clear prompts; as the authors show figure 1 as a motivation, it is still necessary to better analyse the two type errors in the final trained model, not like figure 3, but like figure 1 to show the improvement.

**Questions:**

See above

---

> ### Author Response · Authors · 2024-11-19
> **Official Response from Authors [1/2].**
>
> We appreciate your valuable time, insights, and for highlighting our strengths (i.e., **the novelty of our ideas**, **the clear steps behind our method**, and **the completeness of experiments**). In the following, we will carefully respond to your questions.
>
> ---
>
> 1. **Question about motivation and clarity in the problem statement.**
>
>    - Thank you for your insights and kind suggestions! We agree with your perspective; our paper indeed focuses primarily on the deficiencies and improvements of open-sourced MLLMs. As you pointed out, proprietary MLLMs like Claude and GPT-4o may be advanced in these ways. In the revised manuscript, we will address this by rephrasing the introduction to more explicitly highlight our focus.
>    - At the same time, we also think that **even these models also have room for improvement, particularly in reasoning**. As shown in Table 4 of our paper, while some proprietary models may excel in recognition, they still fall short in reasoning-intensive tasks compared to humans (e.g., the performance gap on ReachQA-Reas. and CharXiv-Reas.). Additionally, IsoBench [1] also demonstrates that even models like GPT-4, Gemini Pro, and Claude 3 Opus achieve more than 15-point advantages with text input over image input, **highlighting opportunities for improvement through cross-modal distillation**.
>    - In conclusion, although we could only validate our approach on open-sourced MLLMs, the consistent improvements across diverse base models (e.g., LLaVA -> MiniCPM -> InternVL2) suggest that our method and dataset have the potential to benefit a broader range of MLLMs, including proprietary ones. We hope the clarified narrative will address your concerns.
>
> 2. **Question about limited novelty in the proposed method/framework, though the idea of CIT is appreciated.**
>
>    - We acknowledge that the CIT framework incorporates several existing techniques, as they are well-established and have been widely validated for their feasibility and effectiveness.
>    - That said, we would like to gently shift the focus toward the proposed concept of “code-as-intermediary” rather than the specific steps of the framework. By translating images into textual representations, **this approach facilitates the seamless integration of various text-based data synthesis methods**—such as Tree-Instruct [2], OSS-Instruct [3], and potentially others—into our workflow. This characteristic enhances the adaptability and scalability of the framework.
>    - Furthermore, we believe that with the advancement of controllable generation techniques, the “intermediary translation” concept could be extended to the synthesis of diverse multimodal data types across various domains.
>
> 3. **Question about the analysis of question types.**
>
>    - Thank you for your suggestion! We intentionally avoid restricting question types during the generation process to enhance its diversity, though this may reduce transparency. In response, we have added question type statistics for 2k ReachQA test set questions, categorized by GPT-4o. The results are as follows:
>
>      | Question Type            | # in Reco. split | Question Type           | # in Reas. split |
>      | :----------------------- | :--------------: | :---------------------- | :--------------: |
>      | Direct Lookup            |       177        | Trend Inference         |       147        |
>      | Comparative Lookup       |       326        | Causal Analysis         |        63        |
>      | Attribute Identification |       123        | Conditional Reasoning   |        55        |
>      | Pattern Recognition      |        61        | Comparative Analysis    |       267        |
>      | Compositionality         |       313        | Quantitative Evaluation |       378        |
>      | Others                   |        0         | Others                  |        90        |
>      | Total                    |       1000       | Total                   |       1000       |
>
>
> ---
>
> ## Reference
>
> [1] IsoBench: Benchmarking Multimodal Foundation Models on Isomorphic Representations, COLM 2024
>
> [2] A Preliminary Study of the Intrinsic Relationship between Complexity and Alignment, COLING 2024
>
> [3] Magicoder: Empowering Code Generation with OSS-Instruct, ICML 2024

---

> ### Author Response · Authors · 2024-11-19
> **Official Response from Authors [2/2].**
>
> 4. **Question about the performance comparison with close models.**
>
>    - We acknowledge that there remains a performance gap compared to closed-source models, which we hypothesize could be due to several factors. One possible reason is the **inherent limited capabilities of the base model we employed**. Additionally, it is unclear whether the closed-source models have been fine-tuned on similar or even more extensive task-specific datasets, which might have provided them with a notable advantage in these benchmarks.
>    - Another point worth emphasizing is that our approach **prioritizes maintaining a balance between task-specific performance and broader generalization abilities**. For example, while some chart-augmented models like ChartLlama [4] and ChartAssistant [5] achieve strong results on specific tasks (e.g. ChartQA) by leveraging significantly larger datasets (**e.g., ChartAssistant uses 39M samples, approximately 2,000× our dataset**), such specialization often comes at the cost of reduced versatility in other areas, as observed in tasks like MathVista.
>    - Despite these constraints, our work demonstrates meaningful progress. As shown in Table 4 of our paper, the fine-tuned InternVL2-8B achieves an average performance level approaching GPT-4o Mini (48.35 v.s. 49.34), highlighting its potential to **achieve competitive and balanced results even with a relatively small dataset and weak model**.
>
> 5. **Question about the simplicity of the self-repair prompt.**
>
>    - We opted for a simple self-repair prompt because around 80% of code generated by GPT-4o passed on the first execution. By appending error messages and self-repair prompting for correction, we raised this success rate to 95% (Line 274, 275). While more complex workflows [6,7] may further improve this result, they would also incur additional costs.
>
> 6. **Question about error analysis in trained models.**
>
>    - Thank you very much for this suggestion! We have added an analysis of errors before and after training to provide a clearer understanding of the improvements. The results are as follows:
>
>      | Model                          |  # Errors  | # Reco. Error | Reco. Proportion | # Reas. Error | Reas.  Proportion |
>      | ------------------------------ | :--------: | :-----------: | :--------------: | :-----------: | :---------------: |
>      | LLaVA-Next-Llama3-8B           |    1355    |      827      |      61.0%       |      493      |       36.4%       |
>      | LLaVA-Next-Llama3-8B + ReachQA | 886 (-469) |  534 (-293)   |    60.3% (↓)     |  334 (-159)   |     37.7% (↑)     |
>      | MiniCPM-V2.5-Llama3            |    827     |      513      |      62.0%       |      297      |       35.9%       |
>      | MiniCPM-V2.5-Llama3 + ReachQA  | 714 (-113) |   414 (-99)   |    58.0% (↓)     |   278 (-19)   |     38.9% (↑)     |
>      | InternVL2-8B                   |    655     |      335      |      51.1%       |      296      |       45.2%       |
>      | InternVL2-8B + ReachQA         | 439 (-216) |  209 (-126)   |    47.6% (↓)     |   198 (-98)   |     45.1% (↓)     |
>
>    - The results demonstrate that: (1) For different models, both types of errors show a noticeable reduction, indicating that **the trained model improves in both recognition and reasoning abilities**. This is consistent with the observations in Sections 4.2 and 5.2 of our paper. (2) Additionally, stronger base models (e.g. InternVL2-8B) exhibit a lower proportion of recognition errors, and after training with ReachQA, the reduction in recognition errors is more significant. This suggests that **recognition, as a more fundamental capability, could benefit more directly from the training**.
>
> ---
>
> ## Reference
>
> [4] ChartLlama: A Multimodal LLM for Chart Understanding and Generation, arXiv.2311.16483
>
> [5] ChartAssisstant: A Universal Chart Multimodal Language Model via Chart-to-Table Pre-training and Multitask Instruction Tuning, ACL 2024
>
> [6] Self-Edit: Fault-Aware Code Editor for Code Generation, ACL 2023
>
> [7] SWE-agent: Agent-Computer Interfaces Enable Automated Software Engineering, NeurIPS 2024

---

> ### Author Response · Authors · 2024-11-29
> **A Friendly Reminder from Authors.**
>
> Dear Reviewer cHNi,
>
> We hope this message finds you well. If this email arrives during your vacation or outside your normal working hours, please accept our apologies for the interruption.
>
> It has been 10 days since our last round of communication. We just want to kindly follow up to ensure that we have addressed your concerns or any remaining questions you might have. We are still here and welcome any further discussion or feedback, as your insights are incredibly valuable to us.
>
> Thank you very much for your time and consideration.
>
> Best regards,
>
> Authors of Submission 5783

---

> ### Author Response · Authors · 2024-12-02
> **Gentle Reminder: Discussion Phase Ending.**
>
> Dear Reviewer cHNi,
>
> We hope this message finds you well. We truly appreciate the time and effort you have already dedicated to reviewing our submission.
>
> As the Author-Reviewer Discussion phase is nearing its conclusion, we just want to kindly follow up one last time. If there are any additional points you’d like us to address or clarify, we are here and ready to respond promptly. Should the issues be resolved, we kindly ask for your consideration in adjusting your scores accordingly.
>
> Thank you again for your time and consideration!
>
> Best regards,
>
> Authors of Submission 5783

---

### Official Review · Reviewer_U7uB · 2024-11-12

**Soundness:** 2
**Presentation:** 2
**Contribution:** 2
**Rating:** 5
**Confidence:** 4

**Summary:**

This paper proposes a novel pipeline for constructing multi-modality chart datasets. The central idea is the introduction of the “Code-as-Intermediary Translation” concept, which facilitates the creation of chart datasets for MLLMs (Multimodal Large Language Models). Experimental results demonstrate that the proposed datasets not only enhance chart-related QA capabilities but also improve overall reasoning performance.

**Strengths:**

Utilizing code as a generative engine is a sound approach to creating new chart images.

Experimental results indicate that the new dataset significantly enhances specific performance metrics.

**Weaknesses:**

The use of code as a tool for rendering the chart dataset appears similar to the concept presented in [1]. Furthermore, data augmentation through code is reminiscent of [2]. Could you elaborate on how your approach differentiates from these works?

The model fine-tunes general MLLM models. How would the performance change if an equivalent amount of chart data were selected from general datasets for fine-tuning the MLLMs? A comparative analysis might be insightful.

Why is the chart type described as ∞? Are there any specific chart types that cannot be generated using Python code?

What is the accuracy or success rate of MLLMs when generating code?

**Questions:**

See Weaknesses

---

> ### Author Response · Authors · 2024-11-13
> **A Friendly Reminder from Authors**
>
> Thank you for your thoughtful comments and insights into our work. We greatly appreciate your recognition of our efforts and the questions raised. However, we have a few follow-up questions for understanding:
> 1. We notice that in your review, **the references marked as [1] and [2] were mentioned but not specifically listed**. Could you clarify or provide the full citations for these references? This would help us better understand the context and address your query more accurately.
> 2. Regarding Weakness 2, we want to seek clarification to ensure we fully understand your concern. From our understanding, you may be suggesting that, as we aim to fine-tune general MLLM models, we should consider comparing chart data from a more general dataset, rather than a specific dataset discussed in Section 5.3. This might involve, for example, sampling an equivalent amount of chart-related data from a broader multimodal dataset like [The Cauldron](https://huggingface.co/datasets/HuggingFaceM4/the_cauldron).
>
> If our understanding is incorrect, we would appreciate any further details you could share, so we can thoroughly address your concern as soon as possible.
> Thank you again for your dedicated review and feedback.

---

> ### Comment · Reviewer_U7uB · 2024-11-18
> **reference**
>
> 1. Sorry for missing the references.
>
> [1]. https://arxiv.org/pdf/2409.01704
>
> [2]. WizardLM: Empowering Large Language Models to Follow Complex Instructions
>
>
> 2. Yes, or other datasets, such as Cambrian-10M.

---

> ### Author Response · Authors · 2024-11-19
> **Official Response from Authors.**
>
> We appreciate your valuable time, insights, and for highlighting our strengths (i.e., **the sound approach** and **the effectiveness of ReachQA**). In the following, we will carefully respond to your questions.
>
> ---
>
> 1. **Question about the use of code for rendering charts and data augmentation**.
>
>    - Thank you for your thoughtful feedback and for highlighting these important references. We acknowledge that our approach shares similarities with GOT [1] in leveraging code for synthetic chart data generation. However, it is worth noting that **using code as a rendering tool is a fundamental and widely adopted practice in chart generation**. It is not unique to our work or GOT [1]; earlier works such as ChartLlama [2] and ChartAst [3] also used code to build datasets for chart-related tasks. The use of programming languages (e.g., Python, R) and libraries (e.g., Matplotlib, Seaborn) is a common choice in this field due to their flexibility and precision in chart visualizations.
>    - The second paper you mentioned, WizardLM [4], is a pioneering work that introduced the concept of Evol-Instruct to enhance data complexity. While they focus entirely within the textual modality, our motivation diverges significantly. Given the current scarcity of visually complex and reasoning-intensive chart datasets, we adopt an innovative approach by **introducing the concept of "code-as-intermediary"—translating images into textual representations and subsequently adapting text-based data synthesis techniques**, including Evol-Instruct (see Section 3.1, Line 247). In summary, while we acknowledge the importance of [4] as an inspiring foundation, our contribution lies in adapting it to a novel use case, using code to bridge visual and textual modalities and tackle the unique challenges of chart-related tasks.
>
> 2. **Question about fine-tuning with chart data from general datasets.**
>
>    - Thank you for your suggestion! To address your point, we conducted additional experiments using an equivalent amount of chart-related data from The Cauldron [5], a massive collection of 50 vision-language datasets designed for general MLLM fine-tuning. Specifically, *The Cauldron*'s chart understanding subset is **a mixture of seven distinct datasets**, and we followed the setting in Section 5.3 of our paper to proportionally sample 20k data for training. The results are as follows:
>
>      | Models         |   Avg.    |  ReachQA  |  CharXiv  | MathVista |  Math-V   |
>      | :------------- | :-------: | :-------: | :-------: | :-------: | :-------: |
>      | Base Model     |   16.39   |   6.50    |   17.20   |   32.40   |   9.44    |
>      | + The Cauldron |   18.61   |   10.10   |   19.10   |   35.60   |   9.64    |
>      | + ReachQA      | **20.74** | **11.10** | **22.50** | **38.10** | **11.25** |
>
>    - The results show that training with this general mixed dataset achieves slightly better performance compared to using individual datasets (e.g., ChartBench, ChartAst). However, **the benefits are limited due to the quality of the included data**, which features relatively simple charts, straightforward questions, and inadequate rationale annotations. As a result, the performance still lags behind fine-tuning with high-quality synthetic datasets like ChartGemma and ReachQA.
>
> 3. **Question about the ∞ chart type.**
>
>    - Sorry for this confusion, but there seems to be some misunderstandings. In Table 1 of our paper, we specify “**# Chart Type: 10 / 32**”, indicating that our method can synthesize 10 major chart types and 32 subtypes, all of which can be generated using Python’s standard or third-party libraries.
>    - The use of “**# Chart Topic: ∞**” reflects that our method does not restrict chart topics. Through a Self-Instruct approach, the model can freely expand on themes based on its knowledge, enabling a wide variety of topics to be represented. If needed, we will update the description in the revised manuscript to ensure clarity.
>
> 4. **Question about the success rate of code generation.**
>
>    - We appreciate your interest in this detail. As Section 3.2 (lines 274–275) mentions, the execution success rate of code generated by GPT-4o is approximately 80%, which improves to about 95% after applying a Self-Repair method.
>    - We also experimented with other models, such as Llama-3.1-70B-Instruct, which initially achieved around a 60% pass rate. However, through iterative Self-Repair, this also improved to over 90%.
>
> ---
>
> ## Reference
>
> [1] General OCR Theory: Towards OCR-2.0 via a Unified End-to-end Model, arXiv.2409.01704
>
> [2] ChartLlama: A Multimodal LLM for Chart Understanding and Generation, arXiv.2311.16483
>
> [3] ChartAssisstant: A Universal Chart Multimodal Language Model via Chart-to-Table Pre-training and Multitask Instruction Tuning, ACL 2024
>
> [4] WizardLM: Empowering Large Language Models to Follow Complex Instructions, ICLR 2024
>
> [5] https://huggingface.co/datasets/HuggingFaceM4/the_cauldron

---

> ### Author Response · Authors · 2024-11-29
> **A Friendly Reminder from Authors.**
>
> Dear Reviewer U7uB,
>
> We hope this message finds you well. If this email arrives during your vacation or outside your normal working hours, please accept our apologies for the interruption.
>
> It has been 10 days since our last round of communication. We just want to kindly follow up to ensure that we have addressed your concerns or any remaining questions you might have. We are still here and welcome any further discussion or feedback, as your insights are incredibly valuable to us.
>
> Thank you very much for your time and consideration.
>
> Best regards,
>
> Authors of Submission 5783

---

> ### Author Response · Authors · 2024-12-02
> **Gentle Reminder: Discussion Phase Ending.**
>
> Dear Reviewer U7uB,
>
> We hope this message finds you well. We truly appreciate the time and effort you have already dedicated to reviewing our submission.
>
> As the Author-Reviewer Discussion phase is nearing its conclusion, we just want to kindly follow up one last time. If there are any additional points you’d like us to address or clarify, we are here and ready to respond promptly. Should the issues be resolved, we kindly ask for your consideration in adjusting your scores accordingly.
>
> Thank you again for your time and consideration!
>
> Best regards,
>
> Authors of Submission 5783

---

### Author Response · Authors · 2024-11-24
**Overall Response from Authors.**

Dear Reviewers,

We are very grateful for your recognition of our work (e.g., the **novelty** of our idea, the **effectiveness** of our method and dataset, and the **completeness** of our experiments), which is a great encouragement to us.

We also appreciate the constructive suggestions, and have made updates to our paper based on your feedback. **All revisions are highlighted in red for your ease of reference**. In this overall response, we provide a summary of the key actions we have undertaken.

---

## Clarifications

1. **The comparison between our proposed CIT and other data synthesis methods, as well as our innovations and contributions.** (Reviewer U7uB & cHNi)
   - Our approach builds upon well-established practices in data synthesis, such as leveraging code for rendering charts and augmenting seed data. While these foundational aspects are shared with prior work, our key innovation lies in introducing the **“code-as-intermediary”** concept, which bridges visual and textual modalities to tackle unique challenges in chart-related tasks. By enabling the seamless application of existing text-based data synthesis techniques within a chart-specific context, our CIT method addresses the current scarcity of visually complex and reasoning-intensive chart Q&A datasets.
2. **The broader applicability of our work beyond open-source MLLMs.** (Reviewer cHNi)
   - While our experiments primarily target open-sourced MLLMs due to accessibility constraints, our evaluation results, along with recent studies like IsoBench, suggest that even advanced proprietary models face challenges in reasoning-intensive multimodal tasks. By leveraging our constructed dataset, we achieve consistent improvements across various base models. This highlights the potential of our approach to enhance the reasoning abilities of open-source or even proprietary MLLMs through effective cross-modal knowledge distillation.
3. **Dataset efficiency and potential for scaling.** (Reviewer 4dEr)
   - Although our current dataset contains only 3k images, our approach effectively leverages the learning potential through carefully designed samples, where each visually complex chart is paired with multiple diverse questions covering different aspects of understanding. Moreover, the automated synthesis pipeline is inherently scalable, making it straightforward to expand the dataset size when needed.

## Supplementary Experiments

1. **Adding a baseline using general (mixed) chart datasets for fine-tuning.** (Reviewer U7uB)
   - We conducted additional experiments using 20k chart-related samples from The Cauldron, a large-scale collection of 50 vision-language datasets chosen for its generality. The results, presented in **Table 5** of the revised paper, reveal that training with this general mixed dataset slightly outperforms using individual datasets. However, it still falls short of the performance achieved by fine-tuning with high-quality synthetic datasets such as our ReachQA.
2. **Adding experiments on two new benchmarks.** (Reviewer Yw4L)
   - We conducted additional evaluations on the OCRBench and We-Math benchmarks to further investigate the generalization of our method. The results, presented in **Figure 4** of the revised paper, align with the trends previously discussed. Specifically, the findings indicate that training on ReachQA enhances reasoning capabilities and incorporating general data yields a well-balanced improvement.
3. **Adding analysis of error types and model improvement.** (Reviewer cHNi & 4dEr)
   - We conducted additional error analyses before and after training with ReachQA to better illustrate the improvements in models. The results are presented in **Appendix C** of the revised paper. This analysis reveals that both recognition and reasoning errors decrease significantly across different models, confirming enhancements in related abilities.
4. **Adding analysis of question types.** (Reviewer cHNi)
   - To demonstrate the diversity and quality of our generated questions, we categorized them into several types using GPT-4o. Detailed statistics are provided in **Appendix D** of the revised paper.
5. **Adding analysis of data contamination.** (Reviewer Yw4L)
   - We conducted an in-depth analysis to address concerns about data overlap. Our comparison included the ReachQA train set against other chart-related benchmarks and its own test set. The results, detailed in **Appendix E** of the revised paper, show our dataset's distinct characteristics from existing benchmarks. To mitigate concerns about data leakage in the test set, we manually reviewed the 50 most similar chart pairs, ensuring no further data contamination was present.

---

Thank you for taking the time to read our response. We hope this addresses your concerns and welcome any further questions or feedback. If our revisions meet your expectations, we kindly ask for consideration in adjusting your scores accordingly.

Best regards,

Authors of Submission 5783

---

> ### Author Response · Authors · 2024-11-27
> **Additional Updates after Author-Reviewer Discussions.**
>
> Here are some additional actions we have undertaken during the discussion phase. We hope these updates further demonstrate the effectiveness of our approach and address the concerns raised. Thank you!
>
> ---
>
> ## Supplementary Experiments
>
> 6. **Adding analysis of qualitative examples.** (Reviewer 4dEr, Round 2)
>    - We conducted a case study visualizing the model's attention patterns to the image during inference, comparing the vanilla model with our fine-tuned version. As presented in **Appendix F** and **Figure 6** of the revised paper, the vanilla model exhibits dispersed attention, leading to redundant outputs and incorrect conclusions, while the fine-tuned model focuses effectively on key elements, enabling accurate and efficient reasoning.
> 7. **Adding detailed results on two new benchmarks.** (Reviewer Yw4L, Round 2)
>    - We conducted more evaluations on the OCRBench and We-Math benchmarks following the setting in Section 5.2 of our paper. Most of the results are consistent with the trends discussed in our paper and the original benchmark papers.

---

### Author Response · Authors · 2024-11-25
**A Friendly Reminder from Authors.**

Dear Reviewers,

As the Author-Reviewer Discussion phase is drawing to a close, we are still here and welcome any further discussion or feedback you may have on our paper. We sincerely look forward to your feedback. Your insights are incredibly valuable to us.

Thank you sincerely for all the time and effort during the review process.

Best regards,

Authors of Submission 5783

---

### Meta-Review · Area_Chair_AED7 · 2024-12-18

**Metareview:**

(a) Summary:
The paper introduces Code-as-Intermediary Translation (CIT), a scalable method for synthesizing chart datasets to enhance reasoning abilities in Multimodal Large Language Models (MLLMs). It contributes the ReachQA dataset, demonstrating improved performance on chart-related tasks and general reasoning benchmarks.

(b) Strengths:
The paper addresses an important gap in MLLM training, proposing a novel and efficient data synthesis pipeline. Extensive experiments show measurable performance improvements, and the dataset exhibits high quality despite its small size.

(c) Weaknesses:
The novelty of the CIT framework is incremental, as it builds on existing techniques. Concerns include limited dataset volume, lack of sufficient qualitative examples, and questions regarding scalability with cheaper models.

(d) Decision:
While the paper is well-executed with valuable contributions, concerns about limited novelty and dataset sufficiency remain. I recommend rejection.

**Additional Comments On Reviewer Discussion:**

During the rebuttal, reviewers raised concerns about dataset size, novelty, and error analysis. The authors addressed these by adding experiments on benchmarks, conducting error analysis, and clarifying distinctions from prior work. While improvements were noted, concerns about scalability and novelty remained significant, leading to the final decision to reject.

---

### Decision · Program_Chairs · 2025-01-22

Reject